# Mechanically interlocked [c2]daisy chain backbone enabling advanced shape-memory polymeric materials

Shang-Wu Zhou[1], Danlei Zhou[1], Ruirui Gu [1]✉, Chang-Shun Ma[1], Chengyuan Yu[1] & Da-Hui Qu [1]✉

The incorporation of mechanically interlocked structures into polymer backbones has been shown to confer remarkable functionalities to materials. In this work, a [c2]daisy chain unit based on dibenzo-24-crown-8 is covalently embedded into the backbone of a polymer network, resulting in a synthetic material possessing remarkable shape-memory properties under thermal control. By decoupling the molecular structure into three control groups, we demonstrate the essential role of the [c2]daisy chain crosslinks in driving the shape memory function. The mechanically interlocked topology is found to be an essential element for the increase of glass transition temperature and consequent gain of shape memory function. The supramolecular host-guest interactions within the [c2]daisy chain topology not only ensure robust mechanical strength and good network stability of the polymer, but also impart the shape memory polymer with remarkable shape recovery properties and fatigue resistance ability. The incorporation of the [c2]daisy chain unit as a building block has the potential to lay the groundwork for the development of a wide range of shape-memory polymer materials.

Certain living organisms in nature can adapt their shapes in response to specific environmental changes and return to their original form when the environment stabilizes. For example, the compound leaves of mimosa pudica fold inward in response to physical touches[1–4]. When insect trespass onto the cavities of Venus flytraps, the plants quickly close their leaves to capture the uninvited guests[5–8]. Chemists have created a variety of synthetic shape-memory materials (SMPs) to mimic this type of biological shape-memory phenomena[9–11]. The reversible shape-shifting processes of the SMPs are usually actuated by temperature[12–15], light[16–19], and other types of external stimuli[20–22]. Triggered by these stimuli, the SMPs alternate between a temporary deformed state and a virginly undeformed state. The force stabilizing the temporary state mostly originates from individual thermal phase transitions[15,23,24], dynamic interactions[14,16,25], and liquid crystal phase transitions[26–28], which can be selectively activated and deactivated. Programmed SMPs can return to their original undeformed state upon actuation, with the mechanism of the shape-memory function involving the storage and dissipation of entropic energy within the polymeric systems[10,11,29,30]. Generally, at the microscopic level, achieving shape-memory behaviors requires the strategic use of non-covalent interactions or chemical bonds[29,31–33].

A mechanical bond is a type of crosslinking that combines the robustness of covalent bonds with the dynamic properties of non-covalent interactions. The mechanically interlocked molecules (MIMs) are molecular architectures consist of two or more compounds that are spatially entangled via mechanical bonds, such as catenanes and rotaxanes[34–38]. The ability to dissociate and associate mechanical bonds from recognition sites has led to the development of stimuli-

[1]Key Laboratory for Advanced Materials and Joint International Research Laboratory of Precision Chemistry and Molecular Engineering, Feringa Nobel Prize Scientist Joint Research Center, Frontiers Science Center for Materiobiology and Dynamic Chemistry, Institute of Fine Chemicals, School of Chemistry and Molecular Engineering, East China University of Science and Technology, Shanghai 200237, P. R. China. ✉e-mail: guruirui@ecust.edu.cn; dahui_qu@ecust.edu.cn

responsive molecular systems that mimic the behavior of biological protein machineries[39–41]. The [c2]daisy chain structure as a topology of mechanically interlocked rotaxane which consists of two half-note-shape subcomponents threading each other, was first synthesized and characterized by Stoddart and coworkers[42]. Sauvage, the pioneer of MIMs, then demonstrated the muscle-like motions of a bistable [c2] daisy chain rotaxane, which aroused the biomimetic research of molecular muscles[43]. Giuseppone and co-workers developed a series of supramolecular materials assembled by pH-sensitive [c2]daisy chain units[44,45]. They further demonstrated that the nanoscopic sliding motions of the interlocked chains can be transformed to macroscopic contractions and expansions[46]. Harada and co-workers introduced photo-responsive azobenzene structures into the backbones of their [c2]daisy chain-based main chain polymers to create dry-type light-driven artificial molecular muscles[47,48]. Yan recently investigated in detail the correlation of [c2]daisy chain backbones and mechanical properties of relevant polymeric materials[49]. These ingenious mechanically interlocked structures bring numerous macroscopic functions when embedded in polymer materials[50–56]. However, the emerge of shape-memory functions by the rational incorporation of mechanically interlocked [c2]daisy chain architectures remains unexplored.

Herein, we develop a thermal-control [c2]daisy chain shape memory (DCSM) polymer by introducing mechanically interlocked [c2]daisy chains to the backbone of polymer networks. The polymer is prepared using a photo-induced thiol-ene click reaction between a four-arm sulfydryl monomer and a mechanically interlocked diene bearing dibenzo-24-crown-8 (DB24C8) rings as the supramolecular hosts and ammoniums as the guests. The resulting SMP exhibits excellent shape fixity ($R_f$) and shape recovery ($R_r$) values, and can be molded into various predetermined shapes. As a proof of concept, the SMP demonstrates its potential as an actuator for releasing and lifting objects. Three control groups are employed to elucidate the role of the [c2]daisy chain topology and the host-guest recognitions in driving the shape memory properties. First, the mechanically interlocked topology of the DB24C8-based [c2]daisy chain contributes to the increase of glass transition temperature which is essential in the entropy-driven shape memory effect. Second, the presence of host-guest interactions within the [c2]daisy chain crosslinks greatly increases the mechanical performance and leads to remarkable shape recovery properties and fatigue resistance.

## Results

### Preparation and characterization of DCSM

As depicted in Fig. 1a, the amorphous functional material DCSM (Supplementary Fig. 1) was created using the olefin-containing monomer DCSM-7 as the precursor of the pseudo-[c2]daisy chains and the four-arm sulfydryl monomer DCSM-SH acting as the junction of the polymer network (Fig. 1a). The half note-like compound DCSM-7 was designed to contain a dibenzo 24-crown-8 (DB24C8) ring connecting with an ammonium-bearing chain. It was synthesized in nine steps (Supplementary Fig. 2) and was characterized by NMR and ESI-mass spectra (Supplementary Figs. 3–10). In polar solvents, the host-guest recognition of crown ether and ammonium is inhibited, causing them to exist as monomers, which is confirmed by the $^1$H NMR spectrum of DCSM-7 in dimethylsulfoxide-d6 (DMSO-d6) (Fig. 1b). Whereas, when this compound was dissolved in nonpolar solvents, such as dichloromethane-d2 (DCM-d2), the peaks of the ammonium protons and the crown ether moiety underwent large variations. The singlet peak of ammonium protons split into two sets of signals and shifted from 8.53 ppm to 6.82-6.84 ppm. The H3 and H4 protons of DB24C8 both shifted upfield and split into two peaks, being consistent with the studies in the NMR characterization of the analogous [c2]daisy chains[57–59]. These changes indicate that DCSM-7 is able to self-assemble via host-guest recognitions to form pseudo-[c2]daisy chain dimers. In

addition, the mass peak of 1437.7524 found in the ESI spectrum also verified the formation of pseudo-[c2]daisy chains (Supplementary Fig. 11).

The functional polymer DCSM was then prepared as a rigid film by the thiol-ene polymerization between the pseudo-[c2]daisy chain and DCSM-SH in DCM under UV irradiation followed by vacuum drying. When the polymerization solvent was substituted with dimethylsulfoxide (DMSO), a vicious liquid was produced after UV initiation, due to the direct covalent linkage of DCSM-7 and DCSM-SH in the absence of a pseudo-[c2]daisy chain structure (Supplementary Fig. 12). On the other hand, when the pseudo-[c2]daisy chain assemblies were added 2 equivalent triethylamine (TEA) before photoinitiation, another viscous liquid was produced due to the disassembly of the pseudo-[c2]daisy chains by deprotonation of the ammonium sites (Supplementary Fig. 13). These two control experiments corroborated that the formation of pseudo-[c2]daisy chains ensure the successful preparation of the polymeric material DCSM (Supplementary Fig. 14).

The formation of polymer networks was further confirmed by swelling experiments and Fourier Transform Infrared Spectrometer (FTIR). The polymer DCSM exhibited different swelling ratios in different solvents (Supplementary Fig. 15), indicating the presence of a covalently cross-linked network in DCSM. The higher swelling ratios of DCSM in less polar solvents (for example, 132.5% in DCM) can be attributed to the hydrophobic nature of the network. The FTIR spectra in Fig. 1c show that the peaks of $v_{=C-H}$ (3080 cm$^{-1}$), $v_{C=C}$ (1640 cm$^{-1}$), and $v_{S-H}$ (2570 cm$^{-1}$) seldom present in DCSM, suggesting the consumption of the monomers. Meanwhile, the $v_{N-H}$ of the ammonium in DCSM-7 shifted from 3439 cm$^{-1}$ to 3403 cm$^{-1}$ contributed by the host-guest interactions between the DB24C8 macrocycles and the ammonium sites. The Energy Dispersive Spectrometer (EDS) mapping was then utilized to ensure the homogeneous dispersion of [c2]daisy chains in DCSM. As depicted in Fig. 1d, all the components, such as phosphorus, fluorine, and nitrogen of the ammonium sites, as well as the sulfur of thiol-ene reaction sites, were uniformly dispersed in the polymer, confirming the creation of a uniform material.

### Shape-memory behaviors of DCSM

Thermal gravimetric analysis (TGA) of DCSM revealed that the decomposition temperature at 95% residue weight is ~300 °C (Supplementary Fig. 16), indicating the material's impressive thermal stability. Differential scanning calorimetry (DSC) detected a glass transition ($T_g$) peak of DCSM at around 40 °C and a broad endothermic peak at around 90 °C associated with the dissociation of host-guest recognition (Supplementary Fig. 17). Similarly, the tan delta curve of DCSM obtained by dynamic mechanical analysis (DMA) also displayed two overlapped peaks at 60 °C (Fig. 2a Fit peak 1) and 90 °C (Fig. 2a Fit peak 2), respectively corresponding to the chain movements attributed by the glass transition and the slide motion of daisy chains. The DCSM could be as a potential shape memory material. The thermo-mechanical behaviors and one-way shape memory behaviors of DCSM were also studied by DMA. As shown in Fig. 2a, the storge modulus of the polymer is larger than its loss modulus from −20 to 140 °C, indicating a good mechanical stability of the DCSM network. The activation energy associated with the slide motion of the DCSM polymer network was calculated to be $349 \pm 4$ kJ mol$^{-1}$ as determined by multi-frequency temperature sweep data (Supplementary Fig. 18). Subsequently, the one-way shape memory cycle of the material was conducted (Fig. 2b). At 90 °C, the polymer DCSM was uniaxially stretched to 35% strain. After cooling to 20 °C, the shape deformation was fixed and the mechanical stress of the instrument simultaneously released. The shape of the polymer recovered to the permanent strain at 90 °C. The shape fixity ($R_f$) and shape recovery ($R_r$) values were respectively 98.8 and 98.6%, indicating the remarkable shape recovery ability of DCSM. Furthermore, the multiple shape-memory cycling of DCSM (Fig. 2c) demonstrated highly consistent reversible actuation

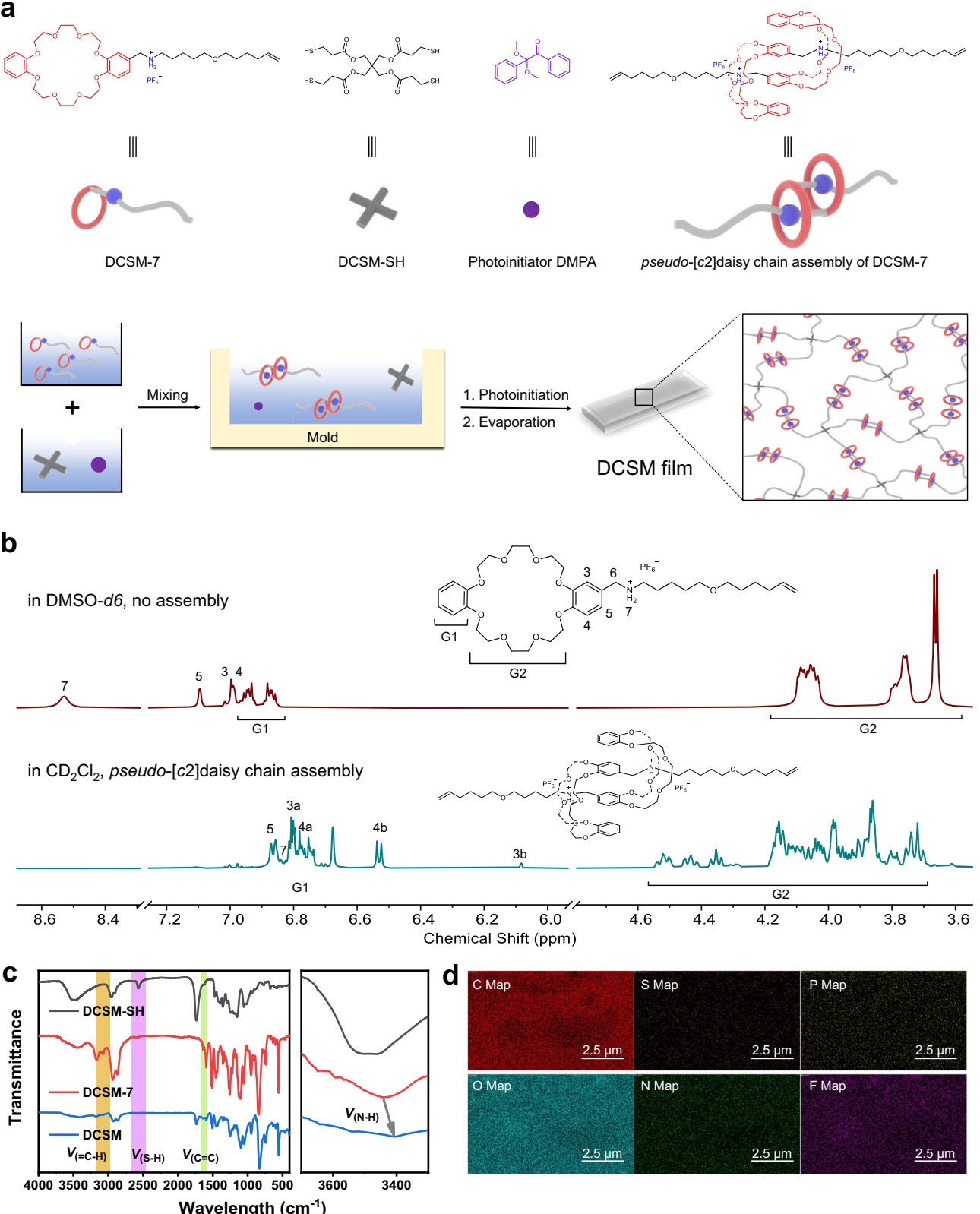

**Fig. 1 | Preparation and characterization of shape memory polymer DCSM. a** Schematic representations of the preparation process and corresponding chemical structures. **b** Partial ¹H NMR spectrum of compound DCSM-7 in DMSO-*d6* and CD₂Cl₂. **c** FTIR spectra of the monomers and the polymer. **d** EDS mapping of a neat DCSM film.

(35% strain) during cyclic heating/cooling between 90 and 20 °C, with no significant change in strain observed after four consecutive shape-memory cycles, highlighting the excellent fatigue resistance of DCSM.

Furthermore, the recovery kinetics of this shape-memory polymer was investigated[60]. A film of DCSM was folded in half to quantitatively monitor the recovery ratio, as indicated by the change in the angle between the folded edges. As shown in Fig. 2d, the actuation

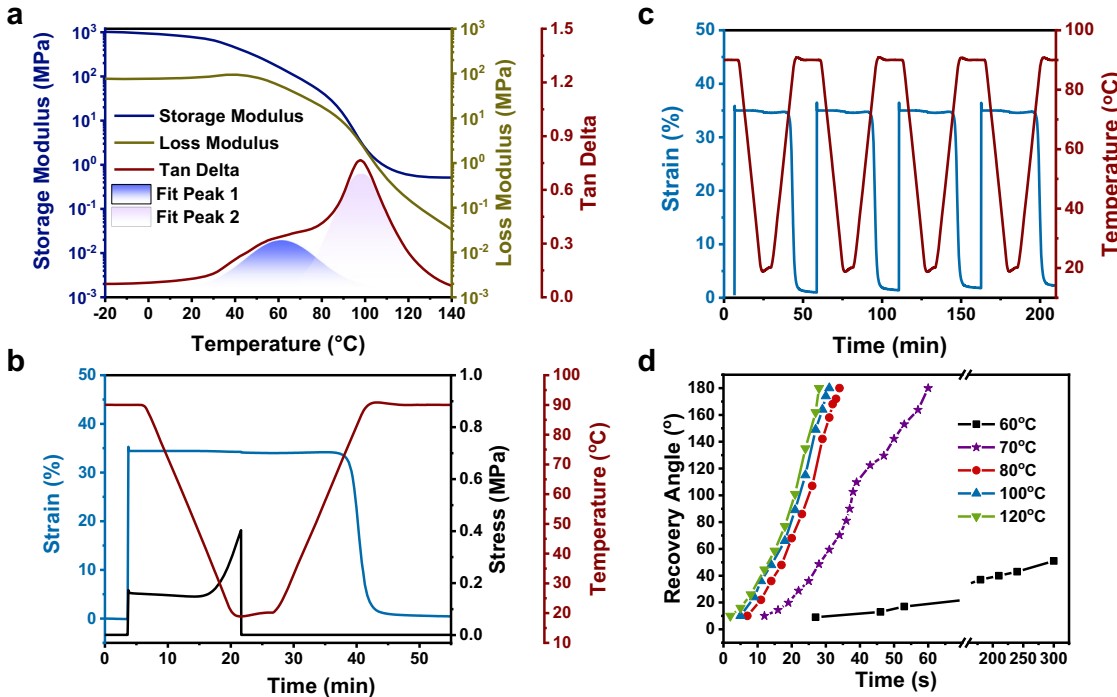

**Fig. 2 | Thermodynamic and shape-memory characterization of DCSM.**
**a** Storage modulus, loss modulus, and tan delta curves of the DCSM polymer obtained by DMA (frequencies: 1 Hz). **b** A shape memory cycle operated by DMA. $R_f = (\varepsilon_d/\varepsilon_{dload})*100\%$ and $R_r = ((\varepsilon_d-\varepsilon_{rec})/\varepsilon_d)*100\%$, with $\varepsilon_{dload}$, $\varepsilon_d$, and $\varepsilon_{rec}$ being the maximum strain under load, the fixed strain, and the recovered strain, respectively. **c** Reversible actuations of DCSM programmed by uniaxial stretching. **d** Recovery kinetics of DCSM under different actuation temperatures.

temperature ($T_a$) had a significant impact on the recovery kinetics. The DCSM film underwent a 100% shape recovery within a short period (<35 s) when the $T_a$ was above 80 °C. This temperature is lower than the DSC endothermic peak (90 °C), suggesting that the shape-memory behavior is almost unaffected by dissociation of the recognition site. The recovery rate significantly slowed down at 70 °C, the acceleration in recovery rate after 40 s came from the addition of gravity in the second half of the recovery process. When the $T_a$ went down to 60 °C, the material recovered with a very slow rate. On the other hand, programming time ($t_p$, the period that the material is held at the folded state before shape recovery) is found to be another parameter that influence the recovery ratio (Supplementary Fig. 19). The recovery rate gradually decreased when the $t_p$ was elongated from 30 to 120 s, while further prolonging $t_p$ did not result in an obvious deceleration of recovery.

To visually demonstrate the one-way shape-memory behavior of DCSM, a polymer film initially shaped as l was repeatedly transformed into temporary shapes representing the letters E, C, U, S and T (Fig. 3a). Upon heating the temporarily shaped film to 90 °C, it reverted back to the l shape (Supplementary Movie 1). Furthermore, this material could retain complex shapes, such as helical and curly forms (Fig. 3b). When these complex-shape materials were programmed into flat strips, they could still be actuated back to the initial states (Supplementary Movie 2 and Movie 3). Similarly, we molded the material into butterfly and blossom shapes to mimic the motions of the corresponding living creatures (Fig. 3c). The mechanical stability and shape memory capability of DCSM indicate its potential in soft robots through programming stimuli-responsive motions. As a proof of concept, the shape memory polymer was programmed to release and lift objects. First, a strip of DCSM was programmed to wrap around a small magnetic stirrer (Fig. 3d). Upon heating in the thermal atmosphere, it released the object (Supplementary Movie 4). When immersed in 90 °C water, the programmed polymer rapidly released the object (Fig. 3d, Supplementary Movie 5). Second, an initial helical DCSM film

hanging two clips was programmed into flat to allow the lifting of the clips (three times its own weight) after heating (Supplementary Movie 6). When the strip is heated in water, it can also lift the objects in 3 s with a lifting distance of more than 1 cm (Fig. 3e, Supplementary Movie 7).

To further investigate the relationship between the mechanically interlocked crosslinker and the macroscopic shape-memory function of the DCSM polymer, the polymer structure is divided into three control groups (see Fig. 4a for their structures). In control group 1 (CG-1), the tetrathiol monomer is treated directly with 1,9-decadiene to form a network without crown ether segments. In control group 2 (CG-2), DB24C8 is incorporated into polymer networks where no recognition site is present (see Supplementary Figs. 20–25 for the characterization data of the synthetic intermediates of CG-2). Control group 3 (CG-3) is designed to be the deprotonated state of DCSM bearing [c2] daisy chain topology in the absence of strong host-guest interactions. These control polymers were prepared as neat films (the detailed preparation protocols are illustrated in the Supplementary Information, see also Supplementary Figs. 26–28 for images of the polymers). Swelling experiments confirmed the formation of crosslinked networks in all control groups (Supplementary Figs. 29–34). CG-3 exhibited a higher degree of swelling in different solvents compared to DCSM, suggesting that the absence of recognition sites results in a more porous network structure, allowing easier solvent penetration. The FTIR spectra of all three control materials indicate the absence of olefin and sulfhydryl peaks, providing additional evidence for the formation of polymer networks (Supplementary Figs. 35–37).

The thermal properties of the three control materials were compared using TGA and DSC. The decomposition temperatures at 95% residue weight were found to be 340 °C, 260 °C, and 250 °C, respectively (Fig. 4b). Their thermal decomposition temperatures higher than 200 °C indicate that the materials have a certain degree of thermal stability. The reduced thermal stability of CG-3 compared to DCSM is likely attributed to the lack of association sites. The DSC curve of CG-1

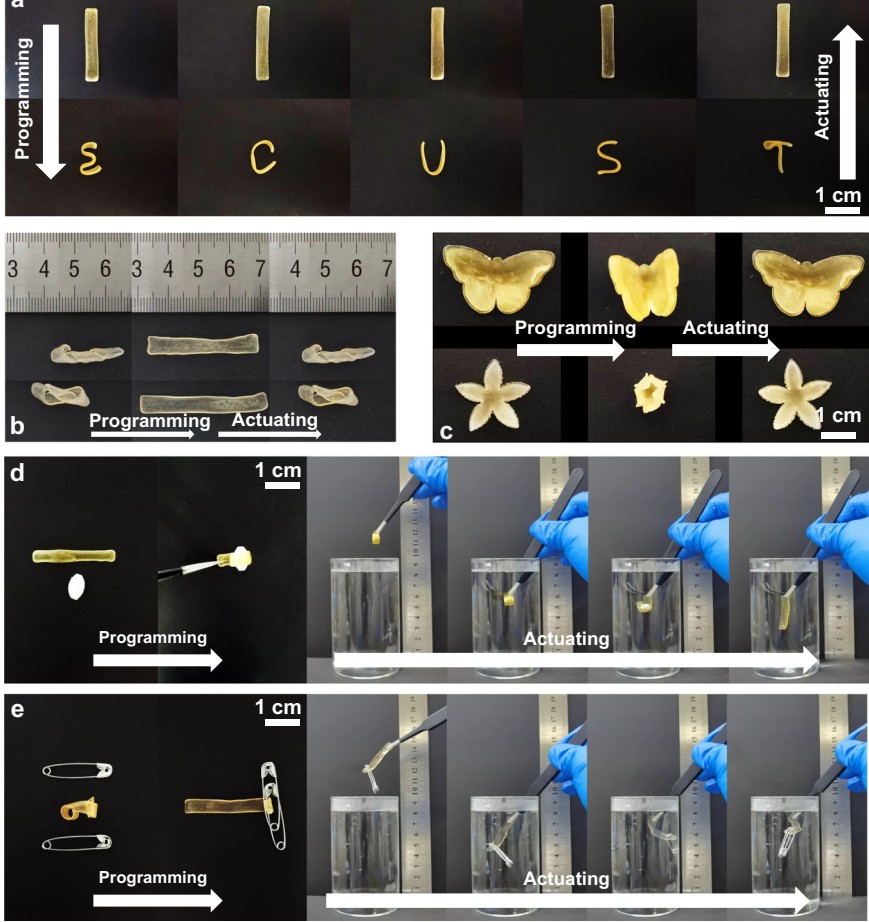

**Fig. 3 | Images of the shape memory behaviors of DCSM. a** A DCSM film programmed to different shapes (letters E, C, U, S, T). **b** Shape memory behavior of DCSM films with initial complex helical and curly shapes. **c** Biomimetic reversible actuation of DCSM polymers with butterfly shape and flower shape. **d** Thermal activated releasing of a stirrer using a programmed DCSM strip ($m_{film}$ = 145 mg, $m_{object}$ = 528 mg) in the air (left) and in the water (right). **e** Thermal activated lifting of two flips using a programmed DCSM strip ($m_{film}$ = 151 mg, $m_{object}$ = 431 mg) in the air (left) and in the water (right). Programming temperature: 90 °C; Actuating temperature: 90 °C.

in Fig. 4c shows a melting peak at 9 °C, which is consistent with the DMA test result (Supplementary Fig. 38). For CG-2, the flexible DB24C8 backbone strengthens the chain segment motion of the polymer, leading to a low $T_g$ at −2 °C (Supplementary Fig. 39). Interestingly, the $T_g$ of CG-3 significantly increased to 32 °C, contributed by the mechanical interlocking structure of [$c$2]daisy chains. In comparison to CG-2, where only one DB24C8 is covalently embedded in the chain segments, the [$c$2]daisy chain segments of CG-3 enhance the network density by the additional DB24C8 rings and spatially restrict the movement of the segments through the mechanical interlocking of the DB24C8 ring. These two effects taken together significantly reduce the free volume of the network and consequently elevate the $T_g$. For DCSM, the additional supramolecular host-guest interactions within the [$c$2]daisy chains further restrict the movement of the segments, resulting in a higher $T_g$ of 40 °C. The tan delta curve of CG-3 also shows two overlapped peaks at 40 and 65 °C (Supplementary Fig. 40). However, both are lower than the corresponding peaks of DCSM due to the weakened ability of energy consumption after deprotonation of the ammonium recognition site[44,61–64].

Interestingly, CG-3 also has a shape-memory function as demonstrated by the DMA test (Supplementary Fig. 41). The shape fixity ($R_f$) and shape recovery ($R_r$) values were respectively 97.6 % and 85.3 %. After 4 cycles, the $R_r$ decreased to 76% (Supplementary Fig. 42), indicating a weaker fatigue resistance compared to DCSM. This is likely due to the flexibility of the [$c$2]daisy chain segments, which are not limited by the location of host-guest complexation. In addition, a CG-3 film could be shaped into a helical form and then reversed through thermal actuation (Supplementary Fig. 43). The multi-frequency temperature sweep tests of CG-3 (Supplementary Fig. 44) yielded an activation energy of 300 ± 7 kJ mol⁻¹, as calculated using the Arrhenius equation (Supplementary Fig. 45), which is ~49 kJ mol⁻¹ lower than that of DCSM, attributed to the enthalpy loss of host-guest interactions. The lower activation energy also results in a lower activation temperature at about 60 °C and accelerated recovery kinetics (9 s at 80 °C) (Supplementary Figs. 46 and 47).

The mechanical properties of the DCSM and the control polymers were tested by uniaxial stretching of their films at room temperature. The resulting stress-strain curves are depicted in Fig. 4d. The control polymers CG-1 and CG-2, being in the elastic state at room temperature, exhibit low tensile strengths of 0.7 MPa and 0.9 MPa, respectively. CG-3 also exhibits a low tensile strength of 1.1 MPa. In comparison, DCSM possesses an elevated tensile strength (14.2 MPa) and Young's modulus (336.2 MPa), indicating a remarkable effect of host-guest interactions within the [$c$2]daisy chains in enhancing the network rigidity (Supplementary Fig. 48). The stress-strain curves of DCSM at different deformation rates were exhibited in Fig. 4e. In addition, we observed a significant dependence of mechanical behaviors of DCSM on the deformation rate. An increase in the strain rate led to a decrease in the fractional strain, indicating the involvement of an energy dissipation mechanism. To trace the possible nanoscopic

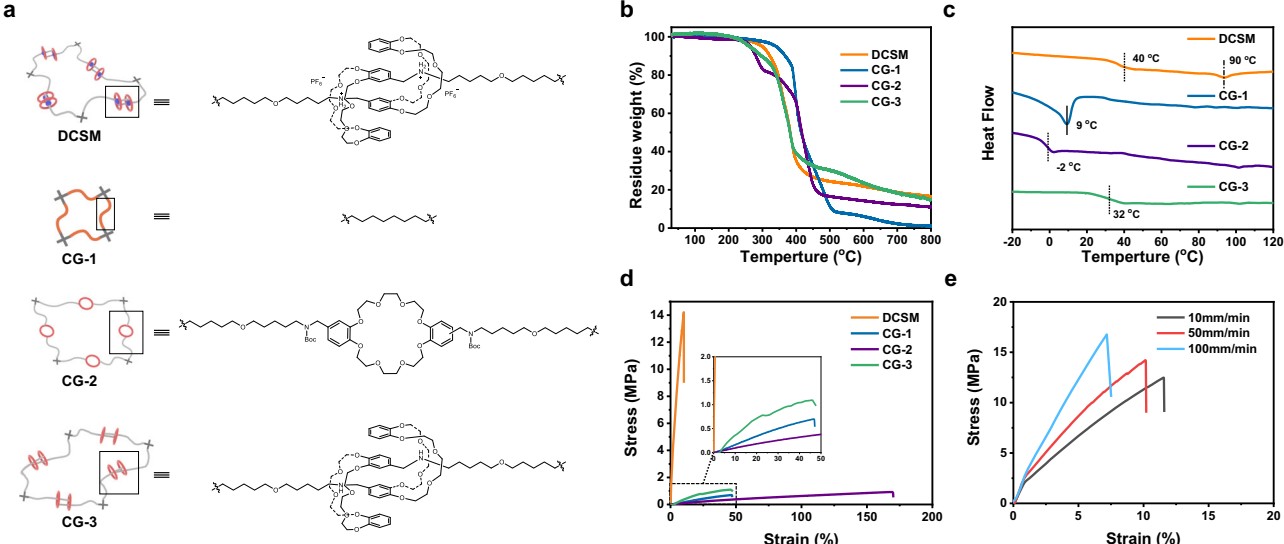

**Fig. 4 | Control groups and characterization of mechanical properties of DCSM and the control groups. a** Schematic representation of the polymer networks of DCSM and its control groups CG−1, CG-2, and CG-3 with molecular structures of their corresponding olefin monomers. **b** TGA curves of DCSM and three control groups from 25 to 800 °C with a heating rate of 20 °C min⁻¹. **c** DSC curves of DCSM and three control groups from −20 to 120 °C with a heating rate of 10 °C min⁻¹. **d** Stress-strain curves of DCSM and three control groups at room temperature (deformation rate: 50 mm min⁻¹). **e** Stress-strain curves of DCSM with different deformation rates (10, 50 and 100 mm min⁻¹).

variations of the polymer network in the shape-memory process, X-ray diffraction (XRD) and small angle X-ray scattering (SAXS) measurements were conducted. However, no distinct XRD diffraction peak (Supplementary Fig. 49) and SAXS scattering peak (Supplementary Fig. 50) were detected for the initial, the programmed, and the recovered DCSM, demonstrating the lack of microscopic orderliness during the shape memory process. Comparison of the FTIR spectra of the initial DCSM and the programmed DCSM shows that the $v_{N-H}$ peak at 3402 cm⁻¹ does not shift (Supplementary Fig. 51), indicating the absence of host-guest dissociation after deformation and the dominance of entropy in the shape memory mechanism. Also, there were no differences in the FTIR spectra of the initial CG-3 and the programmed CG-3 (Supplementary Fig. 52).

Based on the data discussed above, we propose an entropy-driven shape memory mechanism of DCSM as follows: the [c2]daisy chains in the polymer network were initially in free conformation. When the DCSM is heated to a temperature above its $T_g$, the mobility of the molecular chains increases, resulting in a soft, rubber-like material. When the external stress was applied to deform the material, the orientation of the polymer chains underwent changes that lead to a new state of the network with low entropy. This entropic energy is kinetically trapped by cooling the material to a temperature below $T_g$, where the onset of a glassy phase limits the molecular mobility of the chain segments. After reheating to a temperature above $T_g$, the recovered mobility of the chain segments leads to the formation of the permanent shape accompanied by the release of the trapped entropy.

## Discussion

In this study, we prepared a shape-memory polymer called DCSM, which contains mechanically interlocked [c2]daisy chains. The switching behavior is achieved through reversible thermal phase transitions above the glass transition temperature. This polymer exhibits excellent shape memory properties, with a shape fixity of 98.8% and a shape recovery value of 98.6%. In addition, the DCSM polymer demonstrates robust mechanical properties, with a maximum stress of 14.2 MPa and Young's modulus of 336 MPa. Through systematic structure-property-performance studies of DCSM and three control groups, we demonstrate the role of the [c2]daisy chain

crosslinks in driving the shape memory function. We found that the mechanically interlocked topology is an essential element for increasing $T_g$ and consequently enhancing the shape memory function. Furthermore, the supramolecular host-guest interactions within the [c2]daisy chain topology not only ensure robust mechanical strength and network stability of the polymer but also provide the SMP with remarkable shape recovery properties and fatigue resistance. This shape-memory DCSM polymer expands the toolbox for designing shape-memory materials and stimuli-responsive smart materials with tailored dynamic properties and mechanical performance.

## Methods

### Materials

Chemicals were purchased from TCI, Adamas-beta®, and Sigma-Aldrich and used without any further purification. Solvents are reagent grade and were dried and distilled prior to use according to standard procedures.

### Compound synthesis

The synthetic details of compounds can be found in Supplementary Information. The molecular structures are determined using ¹H NMR, ¹³C NMR spectroscopies, and high-resolution electronic spray ionization (ESI) mass spectrometry.

### Preparation of DCSM

Compound DCSM-7 (300 mg, 0.38 mmol) was dissolved in anhydrous $CH_2Cl_2$ (1 ml) in a 5 ml vial and stirred for 30 min at room temperature. DCSM-SH (46 mg, 0.094 mmol) and photoinitiator (2,2-dimethoxy-1,2-diphenylethanone, 5 mg, 0.019 mmol) dissolved in anhydrous $CH_2Cl_2$ (0.5 ml) were then added to the mixture and the solution was allowed to be stirred for 10 min. This precursor solution was poured into a teflon mold (Length * Width * Depth = 80 mm * 6 mm * 1 mm). The mixture was then irradiated under ultraviolet light (50 mW cm⁻²) for 15 min. The obtained film was dried under vacuum overnight at 70 °C. The strip shape can be obtained by direct photopolymerization, and curly and helical polymers can be obtained by chance during photopolymerization. The shape polymerization of butterfly and bud can be obtained by direct photoinitiation in the corresponding mold.

**Fig. 5 | Schematic representation for the kinetic tests of shape-memory processes.** The recovery kinetics of the folded strip was monitored by the folding angle θ as a function of time.

Preparation details for control polymers CG-1, CG-2, and CG-3 can be found in Supplementary Information.

## Kinetic tests of shape-memory processes

A SMP strip (DCSM or CG-3) was placed on a 90 °C heating panel for 30 sec, bent in half, and secured with a 50 g weight (Fig. 5). This programmed strip was then cooled to room temperature for 120 s. Afterwards, the strip was heated to a specified temperature on a heating panel. The recovery process was recorded using a digital phone (Huawei P40Pro). The angle change was further measured by Image J software.

## Shape memory tests of DCSM in different shapes

DCSM polymers in different permanent shapes (strip, helical shape, curly shape, butterfly shape, and flower shape) were programmed into temporary shapes on a 90 °C heating panel for 60 s. The programmed strips were cooled to room temperature for 120 s. Finally, these strips were placed back onto the 90 °C heating panel to perform shape memory processes. The pictures and movies were recorded using a digital phone (Huawei P40Pro).

## Releasing and lifting of objects

DCSM polymers were programmed to their temporary shapes to wrap the corresponding objects on a 90 °C heating panel. They were then grasped by tweezers and immersed into 90 °C water (or heated to 110 °C by a heat gun) to release (or lift) the objects.

## Data availability

The authors declare that the data supporting the findings of this study can be found in the paper and Supplementary Information files, or are available from the corresponding authors upon request.

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

## Acknowledgements

This work was supported by the National Natural Science Foundation of China (grant No. 22025503, 22205064, 22220102004), Shanghai Municipal Science and Technology Major Project (grant No. 2018SHZDZX03), the Fundamental Research Funds for the Central Universities, the Programme of Introducing Talents of Discipline to Universities (grant No. B16017), Science and Technology Commission of Shanghai Municipality (grant No. 21JC1401700), the Starry Night Science Fund of Zhejiang University Shanghai Institute for Advanced Study (grant No. SN-ZJU-SIAS-006), Shanghai Pujiang Program (grant No. 22PJ1402200), the Innovation Program of the Shanghai Municipal Education Commission (2023ZKZD40).

## Author contributions

D. Qu and R. Gu supervised the experiments. S. Zhou conceived the idea. S. Zhou carried out the synthesis and some characterizations of the materials. D. Zhou and C. Ma synthesized some chemical intermediates. R. Gu and C. Yu helped with the manuscript preparation. The manuscript was written by S. Zhou, R. Gu and D. Qu. All authors discussed the results and commented on the manuscript.

## Competing interests

The authors declare no competing interests.
