## [Peer Review File · Nature Communications]

Mechanically Interlocked [c2]Daisy Chain Backbone Enabling Advanced Shape-Memory Polymeric MaterialsEditorial Note: Parts of this Peer Review File have been redacted as indicated to remove third-party material where no permission to publish could be obtained.

REVIEWER COMMENTS

Reviewer #1 (Remarks to the Author):

Shape memory polymers (SMPs) which exhibit promising prospects in various advanced engineering fields are of continuous interests. In the previous reports, the physical interactions or chemical bonds were widely used in the construction of SMPs, while it was generally hard for these decent strategies to avoid some issues like the uncontrollable shape variation or poor processability. In this manuscript, Prof. Qu and co-workers opened up a novel perspective, namely, utilizing mechanical bonds to actuate the shape-memory behaviors. In specific, the authors incorporated the mechanically interlocked [c2]daisy chain units to the polymeric backbone by thiol-ene click chemistry to develop a novel thermal-control SMP. The resultant SMP could be edited into various programmed shape with shape fixity (R_f) of 98.8 %, and was also capable of restoring rapidly to its original shape via controlling the temperature with shape recovery (R_r) of 98.6 %. Impressively, such outstanding shape memory ability allowed the mechanically interlocked SMP serving as an actuator to release and lift heavy objects. Moreover, through a series of controlled experiments, the authors confirmed that [c2] daisy chain junctions in the polymeric backbone acted a crucial role in the shape-memory mechanism of the polymer. Their presence would greatly influence the glass transition temperature and result in the reversible thermal phase transition characteristic of the polymer, therefore bringing about the good shape memory capacity. These exquisitely mechanical interlocked SMP materials would offer broad opportunities for the development of intelligent materials. Therefore, I am pleased to recommend the publication of current work in *Nat. Commun.* after some revisions:

1. In the preparation of the material, the authors utilized FTIR to confirm the formation of the crosslinked network. They found the disappearance of the peak for olefin C–H stretching vibration at 3080 cm^{-1} to indicate the consumption of olefin groups. Actually, the olefin also had the other important FTIR signal at about 1650 cm^{-1} assigned to the C=C stretching vibration. Therefore, I suggest the authors had better add the analysis of this peak in Figure 1c and S25-S27.
2. In Figure S15, the authors also exploited the ^1H NMR spectra of DCSM in CD_2Cl_2 to certify the formation of the crosslinked network. In theory, once be crosslinked, the polymer seems hard to be dissolved in the solvent but exhibits swelling behavior (Otsuka, et al., *Angew. Chem. Int. Ed.* 2023, 62, e2022164). As a result, I speculate the liquid ^1H NMR of Figure S15 might only reflect the structure belonging to a small amount of the soluble oligomeric component of DCSM, while be hard to the whole real structure of crosslinked DCSM. The residual proton signals assigned to olefin in Figure S15b might support the above deduction. Therefore, I suggest it might be more rational for the authors to use solid NMR or swelling experiments to replace the liquid NMR for proving the crosslinked network.
3. In Figure 2b, the authors calculated the activation energy of the glass transition for DCSM with the value of about 42 kJ/mol through the multi-frequency temperature sweep tests. And once the recognition site of [c2]daisy chain is removed, this activation energy would be reduced to about 0.013 kJ/mol. I am curious about the structural reason leading to the great difference and whether the distinguished activation energies would be an important factor to provide the favorable shape-memory behavior of DCSM.
4. In Figure 4c, the authors used the cartoon scheme to illustrate the shape-memory mechanism of the

[c2]daisy chain polymer in a direct view. In this whole process, I notice the crown ethers seem to be always anchored in the recognition site. It is well known that both the applied force and high temperature might drive the crown ether dissociating from the recognition site and sliding along the molecular chain (De Bo, et al., J. Am. Chem. Soc. 2019, 141, 15879-15883). Therefore, I deeply wonder whether the potential force or thermal-induced intramolecular motions of the [c2]daisy chain in the polymeric backbone would have chance to participate in the fascinating thermal-actuated behaviors of DCSM. If so, what role might such intramolecular motions play in the shape-memory ability of DCSM?

5. In the part of references, I notice that some titles of the cited literature have only the first word with capitalized first letter, the others have all words with capitalized first letter. I suggest the authors unify their format.

Reviewer #2 (Remarks to the Author):

This manuscript presents a new polymer chemistry with shape memory capabilities. The authors provide a thorough characterization of the polymer. However, the significance and impact of this work is not clear. The presented polymer does not have unique properties relative to previously developed shape memory polymer systems, particularly considering its complex synthesis route. Beyond concerns about the overall impact of this work, the following should be addressed prior to publication:

1. English language editing is recommended.
2. All acronyms should be defined at first mention in the manuscript (e.g., DCSM, DCM, DMSO).
3. Figure 2a- The tan delta curve seems to show two peaks that are overlapping rather than one broad peak as described in the text. Please provide an explanation for this effect.
4. Figure 2e- Were any temperatures assessed between 60 and 80 degrees C to determine the point at which the change in recovery rates is affected by actuation temperature?
5. Figure 2f- A hypothesis should be provided for why this effect was observed with references to the literature as appropriate.
6. This statement is incorrect, as inhibition of chain motion results in higher T_g, not lower: "For CG-2, the flexible DB24C8 moiety inhibited the chain segment motion of the polymer network to some extent, so it has a low T_g at -2 °C,..."
7. It may be useful to present Figure 4 before Figure 2, as DSC and tensile testing data is ideally used to determine conditions for shape memory testing.

Reviewer #3 (Remarks to the Author):

The group of Qu reports the formation of polymer materials incorporating [c2]daisy chain backbone and claims their shape-memory properties upon thermal stimulation. The article details the synthesis of the [c2]daisy chain architecture, the synthesis of polymer materials including control materials as well as

their thermal and mechanical characterization. Although this work could represent the first example of shape memory polymer including [c2]daisy chain units, I am not sure about the shape-memory properties of the materials, in particular regarding experiments proposed on bulk materials using programming and actuating events (see my comments below for Figure 3 and explanations on page 11). Furthermore, I noticed several missing information and wrong statements which preclude publication at this stage.

General remark: We miss scale bars for all images. They must be added.

Comments on the manuscript:

*Page 4: Reference 49 is not related to a [c2]daisy chain but to a [2]rotaxane and thus, should not be cited there.

* Page 5:

- The preparation of functional polymer DCSM is described as mixing two different DCM solutions, one with the [c2]daisy chain and one with the four-arm monomer. However, the protocol from the supporting information (page 6 of the SI) just mention one DCM solution.
- Protocols for the polymerization in DMSO and with triethylamine before photoinitiation are not reported in the SI. Just some figures described the sample preparation. They should be provided.

* Page 7:

- The authors states that the ^1H NMR spectrum of polymer DCSM (supplementary Fig 15) clearly show the consumption of the alkenyl end groups of the reactants. However, looking at this figure, we clearly see the presence of alkenyl groups at ~ 5.0 and 5.8 ppm, meaning that all terminal alkene units have not been reacted and that the polymerization is not complete after 15 mins of photopolymerization.
- The authors describe FTIR spectra of the different monomers and the resulting polymer but they do not comment on the disappearance of the large band at $\sim 3500\text{ cm}^{-1}$ from DCSM-SH. Could they comment on that?
- Regarding DMA analysis of the materials, the authors claim that the DCSM polymer has a wide range T_g as $\tan(\delta)$ shows a broad peak between 40 and 100°C . However, the glass transition temperature determined using $\tan(\delta)$ is $\sim 100^\circ\text{C}$ as it corresponds to the maximum of $\tan(\delta)$. On the other hand, the T_g determined from the loss modulus is around 40°C (maximum of loss modulus). Such differences are normal as they relates to different properties of the polymers but the description of the DMA analysis should be more rigorous, in particular regarding the subsequent analysis of the DMA experiments which deals with the determination of the activation energy of the glass transition.

* Page 8:

- The authors describe the determination of the activation energy of the glass transition from the DMA analysis at different frequencies. This is a commonly used method to determine this activation energy. However, from the data point in the inset of Figure 2b, I agree that the slope of the Arrhenius plot is around 40 . However, this slope is equal to E_a/R , thus the activation energy is around 330 kJ/mol , which is one order of magnitude higher than the one claimed in the article.

The authors can refer to this article for the measurement:

<https://www.sciencedirect.com/science/article/pii/S1359836806000825>.

- The authors claim that “the low activation energy benefits the material to undergo glass transitions”. I am not sure about the meaning of this sentence but materials with high activation energy such as epoxy materials can also undergo glass transitions.

- The authors investigate the one-way shape memory cycle of their materials using DMA. Here several questions arise. Why the programming temperature is set at 90°C for Figure 2d and at 100 °C for Figure 2e and 2f? How this programming temperature was determined? What happens if the programming temperature is set at 70 or 80°C? Then, the sample is stretch to 35% strain. What is the reference for this 35% strain? How was it determined?

- The authors then reports some strain recovery kinetics at different actuation temperatures and programming times. We do not have any protocols for these measurements. What are the different data corresponding to? How the “angle change” have been measured? Regarding the broad readership of the journal, it would be nice to provide better explanation of this set-up.

* Page 9: Figure 2a: Please precise the frequency at which the DMA analysis was performed.

* Page 11:

- The authors report several shape memory experiments. However, for all of them, we do not have a precise protocol.

Regarding the shape memory experiments described in Figure 3a and 3b, and based on reference 15 from the article, the actuation experiment should show how the programmed shape can be reversibly recovered upon upon heating and cooling cycles. In addition, video of at least one cycle should be provided. **This is, from my point of view, the most critical point of the paper as it really proves the shape-memory effect of the polymer materials.**

For experiments from Figure 3b, can the authors explain how the DCSM materials were molded into complicated shapes. In particular, do they have a mold to make the helical and curly forms?

Regarding experiments from Figure 3d and 3e, and as we do not have any protocols for shape-memory experiments, I am wondering why the authors need to perform the experiments in hot water and why not by just heating the material. Indeed, water can have an influence on the polymer network and thus, the effect observed can be due to water and not to the temperature. In addition, for experiments from Figure 3e, the authors claim a lifting distance of more than 1.5 cm. From the movie, it is very difficult to conclude on this value. Can the authors explain how they performed this measurement?

- The authors describe three control materials. Although they provide NMR for the formation of the polymer network CG-3, we do not have such information for CG-2 and CG-2. Can the authors provide such analysis?

* Page 12: The authors claim that “only DB24C8 moieties of monomer DDCSM-6 interpenetrated the covalent network of CG-2”. I do not understand what is interpenetrated in this CG-2 network. Can the authors comment on this point?

* Page 13:

- The authors determine the glass transition temperature of their materials (CG-1, CG-2, CG-3 and DCSM) using DSC experiments. This is indeed another method to determine T_g. However, why they did not perform the T_g analysis using DMA? In addition, we missed experimental informations regarding the DSC experiments. How many heating and cooling cycles have been performed? Are the data corresponding to the 1st cycle? It would be nice to see the behavior of the materials in both heating and cooling directions.

- For CG-1, why the endothermic peak at 9°C is not considered as a glass transition? The presence of a peak do not preclude the presence of a glass transition (see <https://www.sciencedirect.com/science/article/pii/S0260877407000258>)

- For DCSM, in addition to the transition at 40 °C, we can also observe a small peak at ~100 °C on Figure 4b. Can the authors comment on this second peak and the relationship with the different values

measured by DMA experiments?

- line 308 page 13: it should be "deprotonation" and not "protonation".

* Page 14:

- The authors indicate that for CG-3, 5 cyclic actuations could be achieved. However, compared to DCSM, we can see a drift in the recovery after each actuation? Can the authors comment on this drift?

- The activation energy calculated for CG-3 is wrong. The axes of Supplementary Figure 35 are inverted. Thus, the activation energy should be very similar to DCSM and not 3 orders of magnitude lower.

- For the recovery experiments performed on CG-3 (Supp Figure 35 and 36), we miss the programming temperature. In addition, the recovery is much faster than DCSM. The authors claim that the faster recovery is due to the lower T_g (only 8°C below). However, experiments are performed at a temperature much above T_g, thus I don't think this is the only explanation for the fastest recovery. In addition, the T_g should have an influence on the temperature at which recovery occurs. Thus, regarding the fast recovery at 60°C (as fast as the recovery at 120°C), I recommend the authors to test the recovery at lower temperature (for instance 40°C).

-XRD and SAXS provide information on variations at the nanoscopic scale not at the "microscopic" scale as stated by the authors.

- line 336 page 14: There were "no" (missing in the sentence) distinct differences in FTIR spectra...

* Page 15:

- The authors comment on the behavior of DCSM (from line 348 to line 361) as illustrated on Figure 4c. However, they do not comment on the behavior of CG-3 which is very similar to DCSM, while being a control experiment of this work. They should in particular comment on the difference between the two materials and in particular the presence or absence of the secondary ammonium station, which is the main difference between DCSM and CG-3.

- The authors provide some mechanical characterization of their materials using stress-strain measurements. It is clear that the mechanical properties of DCSM outperform the ones of the control materials. This can be understood for CG-1 and CG-2. However, why the mechanical properties of CG-3 are so different from DCSM? In fact, the stress-strain curves are much more similar to CG-2 than to DCSM. Can the authors comment on that?

* Page 16:

- "The maximum strain (here it should be stress) and Young's modulus gradually increased with the stretching rate, and the maximum stress (here it should be strain) gradually decreased with the stretching rate (Fig. 4f)".

Supporting information:

* Page 6 and 7:

- For all the protocols for the preparation of the DCSM, CG-1, CG-2 and CG-3 polymers, it would be nice to have the number of moles of each reactant, rather than just the quantity. This is important for the ratio between the different monomers.

- The Teflon mold has a volume of 0.5 mL and the precursor solution is around 1.5 mL. Does this mean that a solution is used to fill several molds?

- Protocol for DCSM polymer to be checked according to earlier comments (solution of DCSM-SH and photoinitiator)

- For CG-1 and CG-2, why the solution is stirred for 30 mins and not 10 minutes to be under identical conditions as DCSM?

- For CG-3, 4 mL are used to solubilize DCSM-7 while only 1 mL when preparing DCSM polymer. Dilution can have an effect on the polymer network. Could the authors comment on that point? In addition, after cross linking, the authors use triethylamine to deprotonate the ammonium stations, but the authors do not mention any washing the remaining salts after deprotonation. This could have an influence on the mechanical properties of the materials. And, indeed, ¹H NMR spectra of CG-3 (Suppl. Figure 28) show the presence of the salts.

* Page 11: Supplementary Figures 10 and 11 corresponds to COSY and ESI spectrum of the [c2]daisy chain. However, we do not have a protocol for the formation of the daisy chain nor a characterization by ¹H and ¹³C NMR.

* Page 12: As mentioned earlier, we miss the protocol for materials from supplementary Figure 12 and 13.

* Page 13:

- The image of supplementary figure 14 show a polymer film with a dimension of ~2.9 * 0.5 cm. This is a very different aspect ratio compared to the mold that was used to make the polymer materials (8.0 * 0.6 cm). Can the authors comment on that? This raise also the question of the size of the materials used for stress-strain measurements as Young modulus are directly related to the size and thickness of the measured samples.

- Supplementary Figure 15: We should have more information on the quantities of reactants used to make the DCSM polymer. In addition, it would be nice to quantify the quantity of remaining olefin in the polymer.

* Page 20: Same comment for Supplementary figure 28 as for Supplementary figure 15.

* Page 29:

- Supplementary Figure 29: The NMR spectra of CG-3 and DCSM polymers are very similar in particular in the L1 and L2 regions, although CG-3 should be deprotonated. I am really wondering if the addition of base to the polymer is sufficient to deprotonate the ammonium.

- Supplementary Figure 30: Please precise the frequency at which the DMA analysis was performed.

* Page 22: Why the strain at which the cyclic actuation of CG-3 is performed (suppl Figure 32) is higher than for suppl figure 31 and for DCSM (Figure 2d in the main text, page 9)?

* Page 23: Supplementary Figure 35 is wrong. 1000/T should range between 2.90 and 3.00 while ln(f) should be between 0 and 2.5. This explain also the very low activation energy.

Reviewer #1

comments:

Shape memory polymers (SMPs) which exhibit promising prospects in various advanced engineering fields are of continuous interests. In the previous reports, the physical interactions or chemical bonds were widely used in the construction of SMPs, while it was generally hard for these decent strategies to avoid some issues like the uncontrollable shape variation or poor processability. In this manuscript, Prof. Qu and co-workers opened up a novel perspective, namely, utilizing mechanical bonds to actuate the shape-memory behaviors. In specific, the authors incorporated the mechanically interlocked [c2]daisy chain units to the polymeric backbone by thiol-ene click chemistry to develop a novel thermal-control SMP. The resultant SMP could be edited into various programmed shape with shape fixity (Rf) of 98.8 %, and was also capable of restoring rapidly to its original shape via controlling the temperature with shape recovery (Rr) of 98.6 %. Impressively, such outstanding shape memory ability allowed the mechanically interlocked SMP serving as an actuator to release and lift heavy objects. Moreover, through a series of controlled experiments, the authors confirmed that [c2] daisy chain junctions in the polymeric backbone acted a crucial role in the shape-memory mechanism of the polymer. Their presence would greatly influence the glass transition temperature and result in the reversible thermal phase transition characteristic of the polymer, therefore bringing about the good shape memory capacity. These exquisitely mechanical interlocked SMP materials would offer broad opportunities for the development of intelligent materials. Therefore, I am pleased to recommend the publication of current work in Nat. Commun. after some revisions:

Reply: Thanks for your positive comments and valuable suggestions to improve the quality of our manuscript.

1. In the preparation of the material, the authors utilized FTIR to confirm the formation of the crosslinked network. They found the disappearance of the peak for olefin C–H stretching vibration at 3080 cm⁻¹ to indicate the consumption of olefin groups. Actually, the olefin also had the other important FTIR signal at about 1650 cm⁻¹ assigned to the C=C stretching vibration. Therefore, I suggest the authors had better add the analysis of this peak in Figure 1c and S25-S27.

Reply: Thank you for your advice. We added the green bands to highlight the C=C stretching vibration at 1650 cm⁻¹ in Figure 1d and Supplementary Fig. 33-35 (Figure number is changed due to addition of new figures in the new version). The relevant corrections and discussions have been highlighted in yellow in the revised manuscript.

2. In Figure S15, the authors also exploited the 1H NMR spectra of DCSM in CD2Cl2 to certify the formation of the crosslinked network. In theory, once be crosslinked, the polymer seems hard to be dissolved in the solvent but exhibits swelling behavior (Otsuka, et al., Angew. Chem. Int. Ed. 2023, 62, e2022164). As a result, I speculate the liquid 1H NMR of Figure S15 might only reflect the structure belonging to a small amount of the soluble oligomeric component of DCSM, while be hard to the whole real structure of crosslinked DCSM. The residual proton signals assigned to olefin in Figure S15b might support the above deduction. Therefore, I suggest it might be more

rational for the authors to use solid NMR or swelling experiments to replace the liquid NMR for proving the crosslinked network.

Reply: Thank you for your suggestions on polymer structure characterization. As you pointed out, the polymer DCSM possess both covalent bonds and mechanically interlocked non-covalent bonds. Such polymers are difficult to dissolve in solvents and often exhibit swelling behavior. To characterize the formation of a crosslinked network, we have followed your advice and conducted swelling experiments. As expected, all the polymers involved (DCSM, CG-1, CG-2, CG-3) are swelled in solvents instead of being dissolved. The detailed protocol has been added in the Supplementary Information (SI). The results are presented in the Supplementary Material. As suggested, because the liquid NMR only exhibits a very small amount of soluble monomeric or oligomeric component of DCSM, we have removed the liquid ¹H NMR spectra of DCSM and CG-3 from the manuscript. Revisions and discussions have been added in the main text.

3. In Figure 2b, the authors calculated the activation energy of the glass transition for DCSM with the value of about 42 kJ/mol through the multi-frequency temperature sweep tests. And once the recognition site of [c2]daisy chain is removed, this activation energy would be reduced to about 0.013 kJ/mol. I am curious about the structural reason leading to the great difference and whether the distinguished activation energies would be an important factor to provide the favorable shape-memory behavior of DCSM.

Reply: We are sorry for our mistake in the calculation of the activation energy, which has also been pointed out by Reviewer 3. We have recalculated the activation energies for DCSM and CG-3, respectively being **349** KJ/mol and **300** KJ/mol. The 49 KJ/mol difference comes from the host-guest interaction of DB24C8 and the ammonium site. Actually, some literatures (J. Am. Chem. Soc. 2023, 145, 1, 567-578; Angew. Chem. Int. Ed. 2021, 60, 16224; Angew. Chem. Int. Ed. 2020, 59, 12139-2146) have already reported that the removal of recognition sites in mechanically interlocked polymers lead to a decrease in activation energy. This decrease in activation energy also leads to a decrease in glass transition temperature. When the recognition sites are removed, the ring can move relatively freely along the rod, increasing the mobility of the polymer chains. This results in a decrease in the glass transition temperature and a subsequent decrease in activation energy, which is consistent with the experimental results. Furthermore, this supramolecular interaction, as we conclude in the new version, “not only ensure a robust mechanical strength and network stability of the polymer, but also endow the SMP with remarkable shape recovery property and fatigue resistance ability”, as indicated by the comprehensive comparison of CG-3 and DCSM.

4. In Figure 4c, the authors used the cartoon scheme to illustrate the shape-memory mechanism of the [c2]daisy chain polymer in a direct view. In this whole process, I notice the crown ethers seem to be always anchored in the recognition site. It is well known that both the applied force and high temperature might drive the crown ether dissociating from the recognition site and sliding along the molecular chain (De Bo, et al., J. Am. Chem. Soc. 2019, 141, 15879-15883). Therefore, I deeply wonder whether the potential force or thermal-induced intramolecular motions of the [c2]daisy chain in the polymeric backbone would have chance to participate in the fascinating thermal-actuated behaviors of DCSM. If so, what role might such intramolecular motions play in the shape-memory ability of DCSM?

Reply: Thank you for the discussion on the content of the article. I absolutely agree with your point that both the applied force and high temperature can drive the crown ether dissociating from the recognition site and sliding along the molecular chain. However, in this research, we found that the dominant driving force for the shape-memory effects is still the storage and release of entropic energy instead of the enthalpy from the sliding motion. There are some reasons. First, the thermal-induced dissociation was evidenced by the broad DSC endothermic peak at 90 °C, but the shape-memory function of DCSM could already be operated at a lower temperature (80 °C, fig. 2e). Moreover, the operations at higher temperatures did not give distinct recovery kinetics suggesting that the shape-memory behavior is almost unaffected by dissociation of the recognition site. Second, the FTIR results (seeing SI Figure 40) showed that the $\nu_{(\text{N-H})}$ of the ammonium has no detectable change during the programming and actuation processes.

5. In the part of references, I notice that some titles of the cited literature have only the first word with capitalized first letter, the others have all words with capitalized first letter. I suggest the authors unify their format.

Reply: Thanks for your careful checks. We have carefully checked the manuscript and corrected the errors accordingly.

Reviewer #2

comments:

This manuscript presents a new polymer chemistry with shape memory capabilities. The authors provide a thorough characterization of the polymer. However, the significance and impact of this work is not clear. The presented polymer does not have unique properties relative to previously developed shape memory polymer systems, particularly considering its complex synthesis route. Beyond concerns about the overall impact of this work, the following should be addressed prior to publication:

Reply: Thank you for the comments. According to the systematic structure-property-performance studies of DCSM and the decoupled control groups, we have now established clear insights into the role of [c2]daisy chain and the inside host-guest interactions in driving the shape-memory effects (see revised main text), which has not been previously demonstrated. We think these insights involving the effect of the unique mechanically interlocked molecules will be very useful for the bottom-up design of materials with tailored properties and for the explorations of complex stimuli-responsive materials with multiple functions.

1. English language editing is recommended.

Reply: Thanks for your suggestion. We have tried our best to polish the language in the revised manuscript.

2. All acronyms should be defined at first mention in the manuscript (e.g., DCSM, DCM, DMSO).

Reply: Thanks for your suggestion. We have checked this point. All acronyms are now defined at first mention.

3. Figure 2a- The tan delta curve seems to show two peaks that are overlapping rather than one broad peak as described in the text. Please provide an explanation for this effect.

Reply: Thanks for pointing this out. It is very important and helpful for us to figure out the role of [c2]daisy chain structure. We have added the relevant discussion to the new version as “Similarly, the tan delta curve of DCSM obtained by dynamic mechanical analysis (DMA) also displayed two overlapped peaks at 60°C and 100 °C, respectively corresponding to the chain movements attributed by the glass transition and the slide motion of daisy chains.”

4. Figure 2e- Were any temperatures assessed between 60 and 80 degrees C to determine the point

at which the change in recovery rates is affected by actuation temperature?

Reply: Thanks for the suggestion. In the new figure 2d, we have included the curve using 70 °C as the actuation temperature. This new curve is the intermediate of the curves of 60 and 80 degrees reflecting the mutation of kinetics at different temperatures.

5. Figure 2f- A hypothesis should be provided for why this effect was observed with references to the literature as appropriate.

Reply: In the new manuscript, we hypothesize that the mechanism of the SMP polymer is the storage and release of entropic energy. The programming process is a thermal-induced organization of the polymer chains which gain entropic energy. Longer programming time shall lead to larger chain movement in the network before the critical point, resulting in a longer time required to recover to the original network form. This experiment protocol was added into the Supplementary Material, the result aligns with the literatures (Chem. Mater. 33, 2046-2053 (2021)). The mentioned figure is now Supplementary Figure 19.

6. This statement is incorrect, as inhibition of chain motion results in higher Tg, not lower: “For CG-2, the flexible DB24C8 moiety inhibited the chain segment motion of the polymer network to some extent, so it has a low Tg at -2 °C,…”

Reply: Thank you for pointing this out. We have corrected this sentence to “For CG-2, the flexible DB24C8 backbone strengthens the chain segment motion of the polymer, leading to a low Tg at -2 °C.”

7. It may be useful to present Figure 4 before Figure 2, as DSC and tensile testing data is ideally used to determine conditions for shape memory testing.

Reply: Thank you for your suggestion. Figure 4 mainly involve the comparison of the four materials. In the revised manuscript, we introduce the DSC data of DCSM before the DMA shape memory tests. Tensile test of DCSM at high temperature (90 °C) show a strain range from 0% to 300%, we chose 35% as an empirical condition in shape memory tests. Similar strain ranges have been reported in the literature (Adv. Mater. 2022, 34, 2201679, Angew. Chem. Int. Ed. 2017, 56, 12599 –12602, Angew. Chem. Int. Ed. 2016, 55, 11421 –11425). Therefore, we want to keep the order of these two figures.

Reviewer #3

comments:

The group of Qu reports the formation of polymer materials incorporating [c2]daisy chain backbone and claims their shape-memory properties upon thermal stimulation. The article details the synthesis of the [c2]daisy chain architecture, the synthesis of polymer materials including control materials as well as their thermal and mechanical characterization. Although this work could represent the first example of shape memory polymer including [c2]daisy chain units, I am not sure about the shape-memory properties of the materials, in particular regarding experiments proposed on bulk materials using programming and actuating events (see my comments below for Figure 3 and explanations on page 11). Furthermore, I noticed several missing information and wrong statements which preclude publication at this stage.

Reply: With excitement and gratitude, we sincerely appreciate your meticulous review and feedback. We wholeheartedly embrace your suggestions and consider them with utmost enthusiasm and importance as we proceed to revise our draft. According to your valuable comments and questions, we have conducted a new round of testing, inspection and thinking on the research work, leading to this version with major revisions. As you mentioned, this is the first example of shape memory polymer including [c2]daisy chain units, and we now have established some insights into the relationship between the mechanically interlocked structure and the shape-memory effects. Hope this version resonates with you.

1. General remark: We miss scale bars for all images. They must be added.

Reply: Sorry for the mistakes. We have added scale bars in all images in the revised manuscript.

2. *Page 4: Reference 49 is not related to a [c2]daisy chain but to a [2]rotaxane and thus, should not be cited there.

Reply: We sincerely thank the reviewer for careful reading. As suggested by the reviewer, we have removed this reference.

3. * Page 5:

- The preparation of functional polymer DCSM is described as mixing two different DCM solutions, one with the [c2]daisy chain and one with the four-arm monomer. However, the protocol from the supporting information (page 6 of the SI) just mention one DCM solution.

Reply: We have revised this part in the protocol in the SI as follows: "The DCSM-SH (46 mg) and photoinitiator (2,2-dimethoxy-1,2-diphenylethanone, 5 mg) were dissolved in anhydrous CH₂Cl₂ (0.5 ml) then added to the mixture and stirred for 10 mins."

- Protocols for the polymerization in DMSO and with triethylamine before photoinitiation are not reported in the SI. Just some figures described the sample preparation. They should be provided.

Reply: Thanks for this suggestion. We have added the detailed protocol in the captions of supporting information Fig.12 and Fig.13.

4. * Page 7:

- The authors states that the 1H NMR spectrum of polymer DCSM (supplementary Fig 15) clearly

show the consumption of the alkenyl end groups of the reactants. However, looking at this figure, we clearly see the presence of alkenyl groups at ~5.0 and 58 ppm, meaning that all terminal alkene units have not been reacted and that the polymerization is not complete after 15 mins of photopolymerization.

Reply: Thanks for pointing this out. In fact, we now realize that NMR is not a good characterization for a polymer network. When this crosslinked polymer network is formed, it is hard to be dissolved in the solvent, the presented NMR spectrum thus only reflect the structure of a very small amount of the soluble oligomeric component instead of the whole polymer. In the new version, we have added swelling tests to display the insolubility, which together with the FTIR result confirm the formation of a polymer network. Meanwhile, the NMR spectra has been removed.

5. - The authors describe FTIR spectra of the different monomers and the resulting polymer but they do not comment on the disappearance of the large band at ~3500 cm⁻¹ from DCSM-SH. Could they comment on that?

Reply: Thanks for pointing this out. This ~3500 cm⁻¹ peak is actually still present in the spectrum of the polymer network with a lower intensity, as shown by the shoulder at 3550 cm⁻¹.

6. - Regarding DMA analysis of the materials, the authors claim that the DCSM polymer has a wide range Tg as tan(δ) shows a broad peak between 40 and 100°C. However, the glass transition temperature determined using tan(δ) is ~100 °C as it corresponds to the maximum of tan(δ). On the other hand, the Tg determined from the loss modulus is around 40°C (maximum of loss modulus). Such differences are normal as they relates to different properties of the polymers but the description of the DMA analysis should be more rigorous, in particular regarding the subsequent analysis of the DMA experiments which deals with the determination of the activation energy of the glass transition.

Reply: Thank you for your suggestions. In the new version, the relate discussion was revised as: “Similarly, the tan delta curve of DCSM obtained by dynamic mechanical analysis (DMA) also displayed two overlapped peaks at 60°C and 100 °C, respectively corresponding to the chain movements attributed by the glass transition and the slide motion of daisy chains.” We think this new statement corresponds nicely to the DSC curve showing a Tg at 40°C and an endothermic peak at ~90°C, respectively reflecting the Tg and the host-guest complexation.

7. * Page 8:

- The authors describe the determination of the activation energy of the glass transition from the

DMA analysis at different frequencies. This is a commonly used method to determine this activation energy. However, from the data point in the inset of Figure 2b, I agree that the slope of the Arrhenius plot is around 40. However, this slope is equal to E_a/R , thus the activation energy is around 330 kJ/mol, which is one order of magnitude higher than the one claimed in the article. The authors can refer to this article for the measurement: <https://www.sciencedirect.com/science/article/pii/S1359836806000825>.

Reply: We feel sorry for our carelessness. We have corrected it and we also feel great thanks for your high professionalism. We have recalculated the activation energy values for DCSM and CG-3, and they were calculated to be 349 KJ/mol and 300 KJ/mol, respectively. The 49 kJ/mol difference could be explained by the host-guest interactions within the [c2]daisy chains. We have revised the relevant figures and content in the manuscript accordingly.

8. - The authors claim that “the low activation energy benefits the material to undergo glass transitions”. I am not sure about the meaning of this sentence but materials with high activation energy such as epoxy materials can also undergo glass transitions.

Reply: Sorry for the unclear sentence. We meant to explain the lower T_g and the lower actuation temperature of CG-3 by its lower activation energy compared to DCSM, which has been added into the revised version.

9. - The authors investigate the one-way shape memory cycle of their materials using DMA. Here several questions arise. Why the programming temperature is set at 90°C for Figure 2d and at 100 °C for Figure 2e and 2f? How this programming temperature was determined? What happens if the programming temperature is set at 70 or 80°C? Then, the sample is stretch to 35% strain. What is the reference for this 35% strain? How was it determined?

Reply: Thank you for your questions regarding the testing conditions. The DSC and DMA test results indicate that the glass transition temperature of DCSM is about 40 °C. Generally, the programming temperature is set 40-60°C higher than the glass transition temperature to ensure facile actuations (Polymers 2022, 14, 1598; Adv. Mater. 2022, 34, 2201679; Adv. Mater. 2019, 31, 1905715), so that we chose 90 °C, for the other two figures (now figure 2d and SI figure 19), the programming temperature are also 90 °C, we are sorry for the mistakes in the first version. The 35 % strain was chosen as an empirical condition in our first trials. No obvious difference was detected when we set 40 % or 50 % as the strain so that we keep this value 35 %. Similar strain ranges have been reported in the literature. (Adv. Mater. 2022, 34, 2201679; Angew. Chem. Int. Ed. 2017, 56, 12599 –12602; Angew. Chem. Int. Ed. 2016, 55, 11421 –11425).

10. - The authors then reports some strain recovery kinetics at different actuation temperatures and programming times. We do not have any protocols for these measurements. What are the different data corresponding to? How the “angle change” have been measured? Regarding the broad readership of the journal, it would be nice to provide better explanation of this set-up.

Reply: Thank you for your questions regarding the testing conditions. A reference (reference 59: Chem. Mater. 33, 2046-2053 (2021)) has been cited in the original manuscript to provide details on the testing conditions. According to your suggestion, we have now provided a detailed protocol of this set-up in the SI.

11. * Page 9: Figure 2a: Please precise the frequency at which the DMA analysis was performed.

Reply: Thanks for the suggestion. We have added the frequency (1 Hz) in the revised manuscript.

12. * Page 11:

- The authors report several shape memory experiments. However, for all of them, we do not have a precise protocol.

Reply: Thank for you suggestion. The detailed protocols have been added in the revised manuscript.

13. Regarding the shape memory experiments described in Figure 3a and 3b, and based on reference 15 from the article, the actuation experiment should show how the programmed shape can be reversibly recovered upon heating and cooling cycles. In addition, video of at least one cycle should be provided. This is, from my point of view, the most critical point of the paper as it really proves the shape-memory effect of the polymer materials.

Reply: Thank for the suggestion. We have uploaded three new videos (Movie 1, Movie 2, Movie 3) as supplementary materials recording the shape-memory cycles upon heating and cooling for Figure 3a and Figure 3b.

14. For experiments from Figure 3b, can the authors explain how the DCSM materials were molded into complicated shapes. In particular, do they have a mold to make the helical and curly forms?

Reply: Thank you for your comments. They are actually not molded. By tuning the angle and distance of photoirradiation in the polymerization step, we produced those complicated shapes by chance.

15. Regarding experiments from Figure 3d and 3e, and as we do not have any protocols for shape-memory experiments, I am wondering why the authors need to perform the experiments in hot water and why not by just heating the material. Indeed, water can have an influence on the polymer network and thus, the effect observed can be due to water and not to the temperature. In addition, for experiments from Figure 3e, the authors claim a lifting distance of more than 1.5 cm. From the movie, it is very difficult to conclude on this value. Can the authors explain how they performed this measurement?

Reply: We used hot water to show a different environment where this material can work. Regarding the doubt “the effect observed can be due to water and not to the temperature”, we perform this experiment in air. By just heating the material, the related shape-memory effect driven actuations could be easily accomplished. We have uploaded two videos showing the movements. Regarding the measure of lifting distance, we have added a ruler next to the cup to provide readers with a more visual representation of the movements of the polymer film (Supplement Movie 4 and Movie 6).

16. - The authors describe three control materials. Although they provide NMR for the formation of the polymer network CG-3, we do not have such information for CG-2 and CG-2. Can the authors provide such analysis?

Reply: As we mentioned above, all the four polymers are not soluble in solvent so that NMR spectra are less meaningful. To confirm the formation of polymer networks, we employed swelling tests instead.

17. * Page 12: The authors claim that “only DB24C8 moieties of monomer DDCSM-6 interpenetrated the covalent network of CG-2”. I do not understand what is interpenetrated in this CG-2 network. Can the authors comment on this point?

Reply: Sorry for the unclear statement for the structures of the control groups. In CG-2, DB24C8 is covalently incorporated into polymer networks where no recognition site is present. For clarity, in the revised version, we have added schematic representations of the networks of the control groups and their structures in Figure 4a (also shown below).

18. * Page 13:- The authors determine the glass transition temperature of their materials (CG-1, CG-2, CG-3 and DCSM) using DSC experiments. This is indeed another method to determine T_g. However, why they did not perform the T_g analysis using DMA? In addition, we missed experimental informations regarding the DSC experiments. How many heating and cooling cycles have been performed? Are the data corresponding to the 1st cycle? It would be nice to see the behavior of the materials in both heating and cooling directions.

Reply: Thank you for your suggestion regarding the DSC testing. DSC is well accepted in determining glass transition temperatures and melting points of polymers. We have added the DMA data of CG-1 and CG-2 in the Supplementary Material. To provide readers with a better understanding of our testing methodology, we have included additional experimental details of the DSC experiments in the supplementary information. We followed the standard DSC characterization approach, where the first heating cycle was used to eliminate internal stresses in the material, and the second heating cycle was used to characterize the thermal properties of the material. Additionally, to ensure the accuracy and reproducibility of the experimental results, we performed a third and even a fourth heating cycle to confirm the data obtained in the second cycle.

19. - For CG-1, why the endothermic peak at 9°C is not considered as a glass transition? The presence of a peak do not preclude the presence of a glass transition (see <https://www.sciencedirect.com/science/article/pii/S0260877407000258>)

Reply: Thank you for your discussion about CG-1. We appreciate the literature you shared, we have now conducted DMA of CG-1 which show a T_g peak at -10 °C. Also, XRD show no diffraction peak for CG-1. Therefore, we now take this DSC peak as a T_g peak.

20. - For DCSM, in addition to the transition at 40 °C, we can also observe a small peak at ~100 °C on Figure 4b. Can the authors comment on this second peak and the relationship with the different values measured by DMA experiments?

Reply: Thanks for pointing this out. The endothermic peak at ~90 °C could be explained by the dissociation of host-guest interactions of the [c2]daisy chain. The DMA test also show two overlapped peaks at 60 and 100°C, corresponding to the Tg and slide motion of the [c2]daisy chain structure. We have added relevant discussion in the main text.

21. - line 308 page 13: it should be “deprotonation” and not “protonation”.

Reply: Thank you for pointing this out. It has been corrected.

22. * Page 14:- The authors indicate that for CG-3, 5 cyclic actuations could be achieved. However, compared to DCSM, we can see a drift in the recovery after each actuation? Can the authors comment on this drift?

Reply: Indeed, comparing to DCSM, CG-3 has an obvious loss in fatigue resistance, we explain this difference by the displacements of its daisy chains in the shape-memory processes due to the slide motions of [c2]daisy chains without host-guest interactions. This comparison of the two materials nicely show the importance of host-guest interactions in strengthening the stability and fatigue resistance of materials. The relevant discussion has been added into the main text.

23. - The activation energy calculated for CG-3 is wrong. The axes of Supplementary Figure 35 are inverted. Thus, the activation energy should be very similar to DCSM and not 3 orders of magnitude lower.

Reply: We are sorry for this mistake. We have re-calculated the activation energy for CG-3 to be 300 ± 7 kJ/mol, 49 KJ/mol lower than that of DCSM, which is now reasonable.

24. - For the recovery experiments performed on CG-3 (Supp Figure 35 and 36), we miss the programming temperature. In addition, the recovery is much faster than DCSM. The authors claim that the faster recovery is due to the lower Tg (only 8°C below). However, experiments are performed at a temperature much above Tg, thus I don't think this is the only explanation for the fastest recovery. In addition, the Tg should have an influence on the temperature at which recovery occurs. Thus, regarding the fast recovery at 60°C (as fast as the recovery at 120°C), I recommend the authors to test the recovery at lower temperature (for instance 40°C).

Reply: Thank you for your suggestions regarding the recovery experiments of CG-3. In the new manuscripts, from the updated fig 2e (70°C curve has been added) and Supplementary figure 46 (40°C and 50°C curves has been added), we know that the mutation temperature of DCSM and CG-3 respectively locates at 70°C and 60°C, which is in line with their difference in Tg measured by DSC. The faster recovery of CG-3 could be explained by the lower Tg and the lower activation energy, which is now recalculated to be 49kJ/mol lower than DCSM. The relevant discussion has been included in the main text.

25. -XRD and SAXS provide information on variations at the nanoscopic scale not at the “microscopic” scale as stated by the authors.

Reply: Thank you for pointing this out. It has been corrected.

26. - line 336 page 14: The were “no” (missing in the sentence) distinct difference in FTIR spectra...

Reply: Thank you for pointing this out. It has been corrected.

27. * Page 15:- The authors comment on the behavior of DCSM (from line 348 to line 361) as illustrated on Figure 4c. However, they do not comment on the behavior of CG-3 which is very similar to DCSM, while being a control experiment of this work. They should in particular comment on the difference between the two materials and in particular the presence or absence of the secondary ammonium station, which is the main difference between DCSM and CG-3.

Reply: Thank for your suggestions, this is indeed very important. In the revised version, we have added discussions on the comparison of DCSM and CG-3 about the effects of the host-guest interactions of the [c2]daisy chain unit in mechanical properties and shape-memory properties of the materials.

28. - The authors provide some mechanical characterization of their materials using stress-strain measurements. It is clear that the mechanical properties of DCSM outperform the ones of the control materials. This can be understood for CG-1 and CG-2. However, why the mechanical properties of CG-3 are so different from DCSM? In fact, the stress-strain curve are much more similar to CG-2 than to DCSM. Can the authors comments on that?

Reply: Compared to that of DCSM, the [c2]daisy chain of CG-3 is not limited by supramolecular recognition. This results in a looser network and lower network density, as well as a lower activation energy. According to another paper J. Am. Chem. Soc. 2023, 145, 567–578, the energy dissipation of a [c2]daisy chain under stretch could be very high (10 nm displacement consumpt 1000 kJ/mol, shown below). This helps our results make sense, that the presence of supramolecular recognition results in a large difference in mechanical properties.

[REDACTED]

29. * Page 16: - “The maximum strain (here it should be stress) and Young's modulus gradually increased with the stretching rate, and the maximum stress (here it should be strain) gradually decreased with the stretching rate (Fig. 4f)”.

Reply: Thank you for pointing this out. We have revised this sentence.

Supporting information:

30. * Page 6 and 7:

- For all the protocols for the preparation of the DCSM, CG-1, CG2 and CG-3 polymers, it would be nice to have the number of moles of each reactant, rather than just the quantity. This is important for the ratio between the different monomers.

Reply: Thank for you suggestion for experiment details. The number of moles of each reactant has

been added in revised manuscript.

31. - The Teflon mold has a volume of 0.5 mL and the precursor solution is around 1.5 mL. Does this mean that a solution is used to fill several molds?

Reply: Yes. A 1.5 mL solution of precursor could fill three molds and produce three polymer films.

32. - Protocol for DCSM polymer to be checked according to earlier comments (solution of DCSM-SH and photoinitiator)

Reply: We have added the detailed protocols for DCSM.

33. - For CG-1 and CG-2, why the solution is stirred for 30 mins and not 10 minutes to be under identical conditions as DCSM?

Reply: Thanks for pointing this out. In the previous version, the stirring time to prepare DCSM was 30 min (for self-assembly) + 10 min (for mixing with another monomer and the photoinitiator), the stirring time of CG-1 and CG-2 was 30 min, which we thought is enough for their mixing because no preselfassembly is needed. We understand this comment to have an identical condition for the materials, so we reproduced CG-1 and CG-2 using 30 min + 10 min, no difference has been found in the consequent materials. The relevant preparation method in the SI has been revised.

34. - For CG-3, 4 mL are used to solubilize DCSM-7 while only 1 mL when preparing DCSM polymer. Dilution can have an effect on the polymer network. Could the authors comment on that point? In addition, after cross linking, the authors use triethylamine to deprotonate the ammonium stations, but the authors do not mention any washing the remaining salts after deprotonation. This could have an influence on the mechanical properties of the materials. And, indeed, 1H NMR spectra of CG-3 (Suppl. Figure 28) show the presence of the salts.

Reply: During the preparation of CG-3, when the solvent volume is below 4 ml, the photo-initiated reaction directly forms a gel state that affect the further protonation step. Therefore, a solvent volume of 4 ml was chosen to ensure that the solution remains after photo-initiation. After deprotonation with triethylamine, insoluble salt particles are formed in the solution. Therefore, before evaporating the solvent to prepare CG-3, we filter out these insoluble salt particles to prevent them from being present in the polymer. Regarding the NMR, according to previous answers, CG-3 belongs to covalent-crosslink polymer, so it is unsuitable for structural characterization using NMR which represents the results of oligomers and some monomers.

35. * Page 11: Supplementary Figures 10 and 11 corresponds to COSY and ESI spectrum of the [c2]daisy chain. However, we do not have a protocol for the formation of the daisy chain nor a characterization by 1H and 13C NMR.

Reply: There have been a deeply studies on the self-assembly of the host molecule DB24C8 into [c2]daisy chain structures.(Chem. Sci., 2016, 7, 1696–1701; Sci. China Chem. 2019, 62, 245-250.) The DCSM-7 assembles into [c2]daisy chains is a spontaneous process that occurs in nonpolar solvents such as DCM. After dissolving DCSM-7, it can assemble into stable [c2]daisy chain structures. The formation of the [c2]daisy chain structure can be fully characterized through NMR and ESI spectroscopy. The protocol has been added in caption of supplementary Figure 10 and 11.

36. * Page 12: As mentioned earlier, we miss the protocol for materials from supplementary Figure 12 and 13.

Reply: As you mentioned earlier, we have added the protocols for materials in the revised manuscript.

37. * Page 13:- The image of supplementary figure 14 show a polymer film with a dimension of $\sim 2.9 * 0.5$ cm. This is a very different aspect ratio compared to the mold that was used to make the polymer materials ($8.0 * 0.6$ cm). Can the authors comment on that? This raise also the question of the size of the materials used for stress-strain measurements as Young modulus are directly related to the size and thickness of the measured samples.

Reply: When the mixture was irradiated under ultraviolet light for 15 min, we could obtain the polymer film as a gel, then this film was dried under vacuum overnight at 70 °C. It is a normal occurrence that polymer films undergo a degree of shrinkage after solvent evaporation and thermal curing processes. As the concentration of the photo-initiated solution remains consistent, the size difference between the prepared film and the mold is minimal. As for the impact of film size on mechanical strength, we believe it does not affect the mechanical strength testing of the material since the three-dimensional dimensions of each film are precisely measured before tensile testing.

38. - Supplementary Figure 15: We should have more information on the quantities of reactants used to make the DCSM polymer. In addition, it would be nice to quantify the quantity of remaining olefin in the polymer;* Page 20: Same comment for Supplementary figure 28 as for Supplementary figure 15;* Page 29:- Supplementary Figure 29: The NMR spectra of CG-3 and DCSM polymers are very similar in particular in the L1 and L2 regions, although CG-3 should be deprotonated. I am really wondering if the addition of base to the polymer is sufficient to deprotonate the ammonium.

Reply: Thank you very much for carefully reviewing my article. This was a big error in our testing: covalently crosslinked polymers typically exhibit swelling behavior in solution rather than dissolution behavior. Therefore, the mentioned NMR figures are not characterization data that accurately reflects our polymers, but reflecting a small amount of the soluble monomeric or oligomeric component of DCSM. We have removed these erroneous data and used swelling experiments to characterize the integrity of the materials in this study.

39. - Supplementary Figure 30: Please precise the frequency at which the DMA analysis was performed.

Reply: We have precised the frequency in supplementary Fig. 40 in the revised manuscript.

40. * Page 22: Why the strain at which the cyclic actuation of CG-3 is performed (suppl Figure 32) is higher than for suppl figure 31 and for DCSM (Figure 2d in the main text, page 9)?

Reply: Thanks for the suggestion. We have retested the cyclic actuation of CG-3 under 35% strain (Supplementary Fig. 42).

41. * Page 23: Supplementary Figure 35 is wrong. $1000/T$ should range between 2.90 and 3.00 while $\ln(f)$ should be between 0 and 2.5. This explain also the very low activation energy.

Reply: We are sorry for the error. We have recalculated the values of the activation energy and made corrections to the original figures and text accordingly.

REVIEWER COMMENTS

Reviewer #1 (Remarks to the Author):

The raised concerns have been well addressed. I have no further comments.

Reviewer #2 (Remarks to the Author):

The authors have made substantial improvements to this manuscript. However, the following still needs to be addressed prior to publication:

1. Results, paragraph 3- the added (highlighted) section with synthesis methods would be more appropriately placed in the methods section.
2. The time point used for the added swelling data should be clarified.
3. Figure 2d- How does the percent recovery ratio correlate with unfolding here? It may be more appropriate to plot the angle between the folded edges vs. time in this chart instead. Furthermore, the figure legend stating that this is 'strain recovery' is inaccurate, since you're measuring unfolding rather than shrinking after straining.
4. Figure 3c- Please center the marking indicating the endothermic peak on DSCM with the valley of this peak. There is also an endothermic peak on CG-2 that is not marked. This should either be marked or recognized. Consider using different marker styles for T_g and T_m endotherms to help easily distinguish between the two.
5. The text indicates that the endothermic peak on CG-2 at 9 degrees C is the T_g. However, it appears that this is a peak that returns to baseline (indicative of melting) rather than an endothermic shift typically associated with T_g. Additionally, the marker is placed at the valley rather than at the mid-point of the inflection. Please address these inconsistencies.
6. Supplementary figure 17- the second marker should be centered on the endothermic valley.
7. Supplementary figures 29, 31, and 33: please specify the time frame for measurements in the legends.
8. Supplementary Fig. 46 legend is missing 40 and 50 degrees C.
9. In general, this use of the word 'histogram' in figure legends is not accurate. Please remove.

Reviewer #3 (Remarks to the Author):

Qu and co-workers provide a modified version of their manuscript entitled « Mechanically Interlocked [c2]Daisy Chain Backbone Enabling Advanced Shape-Memory Polymeric Materials». In order to address most of the referees' comments, they have refined their analysis of the data presented in the initial version and performed a series of new experiments. Overall, I think that the new version of the

manuscript strengthens the message of their article. However, I still have some minor comments on their manuscript as listed below, which should be addressed before publication.

- As point out by reviewer 2, the article still requires English language editing.
- Page 4, line 88: The authors claim that the establishment of shape-memory function by the use of mechanical interlocking structures remains unexplored. Although this is true for [c2]daisy chain architectures as said in my initial report, this has already been reported for polypseudorotaxanes (see RSC Ad 2014, 4, 17156). The authors should then change their sentence accordingly.
- Page 5, lines 122, 124 and 136: when DCSM-7 forms dimers, it reaches pseudo-[c2]daisy chain and not daisy chain. This is very important as the dimer of DCSM-7 before polymerization is not yet a fully mechanically interlocked structure. The authors should modify that in order to make it clear for the reader.
- Page 7, lines 154 to 157: The authors report new swelling experiments in different solvents to demonstrate the formation of polymer network. However, they do not comment on the different swelling of the network depending on the solvent, this is also important as it is directly related to the nature of the network. I suggest them to add some comments on that in their manuscript.
- Page 14, line 337: The sentence “leading to an extended form of the [c2]daisy chains”. This sentence is very misleading for people not familiar with molecular machines. In fact, regarding the structure of the daisy chain, the material is almost in its fully extended state when it is produces. Furthermore, in this work, there is no proper actuation of the daisy chain to provide the shape memory properties. As mentioned in the conclusion, the presence of daisy chain mainly influences the T_g and the mechanical strength of the material. Thus, I suggest the authors to modify this sentence accordingly.
- Page 15, line 346: reversible thermal phase transitions are performed above the glass transition temperature, not at the glass transition temperature. Actuation experiments are performed at 90°C while the T_g is at 40°C. The authors should modify this sentence.
- Scale bars are still missing for Supplementary figures 15, 30, 32 and 34.
- The authors indicate that angle changes for strain recovery kinetics have been recorded by “a digital phone”. I was wondering if they used any software such as, for instance, Image J to monitor the angle change recorded with their phone. If so, this information should be included in the SI. If not, they should indicate how they measure the angle from their recording.
- The authors explain that helical and curly forms have been obtained serendipitously by tuning the angle and the distance of photoirradiation. In order to repeat such experiment, a protocol should be provided so that similar forms can be achieved.
- For experiments from Figure 3d (supplementary movie 4), they claim that they use a “thermal atmosphere”. From the movie, we can conclude that they use a heat gun to perform this experiment. Can they precise the temperature at which this experiment was achieved? In particular, regarding the difference of kinetics between the experiment in thermal atmosphere and in water (almost 2 times faster in thermal atmosphere).
- Regarding Figure 3e and supplementary movie 6, I don't think that the authors can claim a lifting distance of more than 1.5 cm. It would be the maximum if we check with the ruler on the size and in fact, it is very difficult to estimate as the strip of polymer is curling.
- For Supplementary figures 35 and 36, we still miss the programming temperature.
- Regarding the difference in stress strain properties between CG-3 and DCSM materials, I am still not convinced by the answer from the authors. First, I don't see how the network density can be different as the starting molecules are almost similar, just lacking the secondary ammonium station. This should not

affect the network density. Second, regarding the article they cite (JACS 2023, 567), as in this article, the authors show another set of experiments starting from the fully contracted state and they reach almost the same relative energy. I would suggest the authors to make at least duplicate of their stress-strain measurements for DSCM and CG-3 so that we can be sure that the information provided on Figure 4d are accurate (We have so far no information on possible duplicate or triplicate of these experiments).

- I thank the authors for modifying the protocols to make the control materials. However, for CG-3, we still miss one important information as indicated in the response to reviewers. All salt particles are filtered before evaporating the solvent. This is very important for repeating this experiment.

- Regarding the shrinkage of the materials I agree with the authors that such phenomenon is common when curing a material. However, I am just surprised by the fact that this shrinkage occurs only in one direction (the longer one) and it occurs by a factor of 2.75. This is very important.

Response to referees:

Reviewer #1 (Remarks to the Author):

comments:

The raised concerns have been well addressed. I have no further comments.

Reply: Thank you for your approval of the changes on the revised manuscript.

Reviewer #2 (Remarks to the Author):

comments:

The authors have made substantial improvements to this manuscript. However, the following still needs to be addressed prior to publication:

Reply: Thank you for the new suggestions. We have made revisions according to them.

1. Results, paragraph 3- the added (highlighted) section with synthesis methods would be more appropriately placed in the methods section.

Reply: Thank you for your advice. We have removed this part to the **Methods** section as a new section after the discussion part in this version. Other important methods have been added from the supplementary information.

2. The time point used for the added swelling data should be clarified.

Reply: Thank you for your suggestion. We have added the swelling time in the revised “Methods for swelling experiments” section and the captions of the corresponding SI Figures.

3. Figure 2d- How does the percent recovery ratio correlate with unfolding here? It may be more appropriate to plot the angle between the folded edges vs. time in this chart instead. Furthermore, the figure legend stating that this is ‘strain recovery’ is inaccurate, since you’re measuring unfolding rather than shrinking after straining.

Reply: Thank you for your suggestion. To be more appropriate, we have modified the recovery ratio Figure to a plot of the angle between folded edges vs. time. Additionally, we have updated the illustration of the angle measurement in the Supplementary Information. In Figure 2d, we have revised the caption. Relevant experimental details have been added to the Methods section.

4. Figure 3c- Please center the marking indicating the endothermic peak on DSCM with the valley of this peak. There is also an endothermic peak on CG-2 that is not marked. This should either be marked or recognized. Consider using different marker styles for Tg and Tm endotherms to help easily distinguish between the two.

Reply: Thank you for your advice. We have centered the marking in the valley of this peak and used different marks in the revised Figure. We think the curve of CG-2 indeed has a small fluctuation but not a peak, probably due to the machine error, plus that in the structure of CG-2 there is no host-guest interaction that can result in an endothermic peak, and DMA of CG-2 did not give a corresponding peak. For these considerations, we did not mark it.

5. The text indicates that the endothermic peak on CG-1 at 9 degrees C is the Tg. However, it appears that this is a peak that returns to baseline (indicative of melting) rather than an endothermic shift typically associated with Tg. Additionally, the marker is placed at the valley rather than at the mid-point of the inflection. Please address these inconsistencies.

Reply: Thank you for your suggestion. We have revised this point. In the new version, we describe this peak as a melting peak and mark this peak at the valley.

6. Supplementary figure 17- the second marker should be centered on the endothermic valley.

Reply: We have revised this marker on the endothermic valley.

7. Supplementary figures 29, 31, and 33: please specify the time frame for measurements in the legends.

Reply: We have added the swelling times in the legends.

8. Supplementary Fig. 46 legend is missing 40 and 50 degrees C.

Reply: Thanks for your careful checks. We have added them in the legend.

9. In general, this use of the word 'histogram' in figure legends is not accurate. Please remove.

Reply: Thanks for your advice. We have removed this inaccurate expression.

Reviewer #3 (Remarks to the Author):

comments:

Qu and co-workers provide a modified version of their manuscript entitled « Mechanically Interlocked [c2]Daisy Chain Backbone Enabling Advanced Shape-Memory Polymeric Materials». In order to address most of the referees' comments, they have refined their analysis of the data presented in the initial version and performed a series of new experiments. Overall, I think that the new version of the manuscript strengthens the message of their article. However, I still have some minor comments on their manuscript as listed below, which should be addressed before publication.

Reply: Thank you for your valuable feedback on our work. They are significant for the improvements of our work. We also appreciate your recognition and approval of the revisions we have made. According to your new suggestions, we now provide this new revised version to meet the requirements for publication.

• As point out by reviewer 2, the article still requires English language editing.

Reply: Thanks for your suggestion. We have re-polished the language in the revised manuscript.

• Page 4, line 88: The authors claim that the establishment of shape-memory function by the use of mechanical interlocking structures remains unexplored. Although this is true for [c2]daisy chain architectures as said in my initial report, this has already been reported for polypseudorotaxanes (see RSC Ad 2014, 4, 17156). The authors should then change their sentence accordingly.

Reply: Thank you for your suggestion. Indeed, the previous statement was not accurate enough. We have cited the mentioned paper as *ref. 56* accordingly. Additionally, we have made the necessary modifications to rectify this inappropriate expression in the manuscript as: “However, the emerge of shape-memory functions by the rational incorporation of mechanically interlocked [c2]daisy chain architectures remains unexplored.”

• Page 5, lines 122, 124 and 136: when DCSM-7 forms dimers, it reaches pseudo-[c2]daisy chain and not daisy chain. This is very important as the dimer of DCSM-7 before polymerization is not yet a fully mechanically interlocked structure. The authors should modify that in order to make it clear for the reader.

Reply: Thank you for your very professional suggestion. Pseudo-[c2]daisy chain is of course more precise here. We have modified the description of the [c2]daisy chain in the manuscript and changed it as pseudo-[c2]daisy chain, which should be clearer for the readers.

• Page 7, lines 154 to 157: The authors report new swelling experiments in different solvents to demonstrate the formation of polymer network. However, they do not comment on the different swelling of the network depending on the solvent, this is also important as it is directly related to the nature of the network. I suggest them to add some comments on that in their manuscript.

Reply: Thank you for your suggestions. We have added new discussions about the swelling experiments in the revised manuscript.

• Page 14, line 337: The sentence “leading to an extended form of the [c2]daisy chains”. This sentence is very misleading for people not familiar with molecular machines. In fact, regarding the structure of the daisy chain, the material is almost in its fully extended state when it produces. Furthermore, in this work, there is no proper actuation of the daisy chain to provide the shape memory properties. As mentioned in the conclusion, the presence of daisy chain mainly influences the Tg and the mechanical strength of the material. Thus, I suggest the authors to modify this sentence accordingly.

Reply: Thank you for your suggestions regarding the statement of proposed shape-memory mechanism. Indeed, there is no clear evidence to suggest that the contraction or extension of [c2]daisy chains has an impact on the shape memory function of the material. We have revised this section to be more precise.

• Page 15, line 346: reversible thermal phase transitions are performed above the glass transition temperature, not at the glass transition temperature. Actuation experiments are performed at 90°C while the Tg is at 40°C. The authors should modify this sentence.

Reply: Thanks for your suggestion. We have modified this sentence.

• Scale bars are still missing for Supplementary figures 15, 30, 32 and 34.

Reply: We have added the scale bars in Supplementary figures 15, 30, 32 and 34.

• The authors indicate that angle changes for strain recovery kinetics have been recorded by “a digital phone”. I was wondering if they used any software such as, for instance, Image J to monitor the angle change recorded with their phone. If so, this information should be included

in the SI. If not, they should indicate how they measure the angle from their recording.

Reply: The angle measurements in the manuscript were performed using Image J software, we have included this detail in the SI.

• The authors explain that helical and curly forms have been obtained serendipitously by tuning the angle and the distance of photoirradiation. In order to repeat such experiment, a protocol should be provided so that similar forms can be achieved.

Reply: Formation of the curled and helical-shaped films occurred accidentally during the light-induced polymerization process. For the helical-shaped film, we selected a light intensity of 50W with a distance of 10 cm between the lamp and the mold. For the curled state of the film, we selected a light intensity of 50 W, with a distance of 15 cm between the lamp and mold. Therefore, we think the distance is an important factor for producing these unexpected shapes. However, we have repeated the experiments but we cannot stably obtain the same shapes. The temperature and humidity can also affect the final shape and thus, for the moment, we still have difficulty in providing a precise and repeatable protocol to form these shapes. Considering that the shape does not affect the shape-memory processes, we did not add related experimental details in the SI. We are still exploring the relationship of thiol-ene polymerization condition with polymer shapes and will present the results in our future articles.

• For experiments from Figure 3d (supplementary movie 4), they claim that they use a “thermal atmosphere”. From the movie, we can conclude that they use a heat gun to perform this experiment. Can they precise the temperature at which this experiment was achieved? In particular, regarding the difference of kinetics between the experiment in thermal atmosphere and in water (almost 2 times faster in thermal atmosphere).

Reply: Thanks for the suggestion. We measured the temperature of the atmosphere heated by our heat gun. The temperature is around 110 °C, 20 °C higher than the heated water, which leads to the faster kinetics of recovery. We have added the temperature to the Supplementary movies 4 and 6.

• *Regarding Figure 3e and supplementary movie 6, I don't think that the authors can claim a lifting distance of more than 1.5 cm. It would be the maximum if we check with the ruler on the size and in fact, it is very difficult to estimate as the strip of polymer is curling.*

Reply: Thanks for your suggestion, we have revised our statement to be accurate. The revised manuscript is "it can also lift objects in 3 seconds with a lifting distance of more than 1 cm".

• *For Supplementary figures 35 and 36, we still miss the programming temperature.*

Reply: We have added the programming temperatures in the legends.

• *Regarding the difference in stress strain properties between CG-3 and DCSM materials, I am still not convinced by the answer from the authors. First, I don't see how the network density can be different as the starting molecules are almost similar, just lacking the secondary ammonium station. This should not affect the network density. Second, regarding the article they cite (JACS 2023, 567), as in this article, the authors show another set of experiments starting from the fully contracted state and they reach almost the same relative energy. I would suggest the authors to make at least duplicate of their stress-strain measurements for DSCM and CG-3 so that we can be sure that the information provided on Figure 4d are accurate (We have so far no information on possible duplicate or triplicate of these experiments).*

Reply: Thank you for your suggestions regarding the mechanical properties. In order to ensure the reliability and reproducibility of the experimental data, we have prepared three batches of DCSM and CG-3 films and have made triplicate of stress-strain measurements. The results obtained (see below) are in line with the experimental data presented in the paper. As for the reference we cited, their structure is a [c2]daisy chain structure with a short sliding distance between the two states (7 atoms between the two stations), much shorter than our structure which have much longer chains (more than 15 atoms) without stoppers. Thus, we speculate that the loss of recognition sites may result in larger displacements of the crown ethers and a significant decrease in the stacking density of polymer chains and further strongly influence the mechanical properties.

• *I thank the authors for modifying the protocols to make the control materials. However, for CG-3, we still miss one important information as indicated in the response to reviewers. All salt particles are filtered before evaporating the solvent. This is very important for repeating this*

experiment.

Reply: Thank you for your suggestions. The salt particles precipitated after the addition of triethylamine to the solution. We then filter the solution before solvent evaporation to prepare the desired polymer film. The details have been added to the SI.

• Regarding the shrinkage of the materials I agree with the authors that such phenomenon is common when curing a material. However, I am just surprised by the fact that this shrinkage occurs only in one direction (the longer one) and it occurs by a factor of 2.75. This is very important.

Reply: Thank you for your interest in the contraction phenomenon. In fact, material shrinkage occurred during the photo-initiation stage. At the beginning of photo-initiation, the material was in a gel state. With increasing photo-initiation time and solvent evaporation effects, the material gradually shrank and ultimately reached the dry-film state. For the moment, we are not sure about the reason of directional shrinkage. As we mentioned in an answer above, the irradiation condition could be a factor that influence the final shapes, which are one of our ongoing research projects. In our future work, we will continue to investigate the reasons behind these interesting phenomena.

REVIEWERS' COMMENTS

Reviewer #2 (Remarks to the Author):

The authors have made substantial improvements to the manuscript and have adequately addressed all reviewer concerns.

Reviewer #3 (Remarks to the Author):

The authors have addressed most of my comments. I there have no more concerns regarding publication.